# Non-Convex SGD Learns Halfspaces with Adversarial Label Noise

**Ilias Diakonikolas**
University of Wisconsin-Madison
ilias@cs.wisc.edu

**Vasilis Kontonis**
University of Wisconsin-Madison
ilias@cs.wisc.edu

**Christos Tzamos**
University of Wisconsin-Madison
tzamos@wisc.edu

**Nikos Zarifis**
University of Wisconsin-Madison
zarifis@wisc.edu

## Abstract

We study the problem of agnostically learning homogeneous halfspaces in the distribution-specific PAC model. For a broad family of structured distributions, including log-concave distributions, we show that non-convex SGD efficiently converges to a solution with misclassification error $O(\mathrm{opt}) + \epsilon$, where $\mathrm{opt}$ is the misclassification error of the best-fitting halfspace. In sharp contrast, we show that optimizing any convex surrogate inherently leads to misclassification error of $\omega(\mathrm{opt})$, even under Gaussian marginals.

## 1 Introduction

### 1.1 Background and Motivation

Learning in the presence of noisy data is a central challenge in machine learning. In this work, we study the efficient learnability of halfspaces when a fraction of the training labels is adversarially corrupted. As our main contribution, we show that non-convex SGD efficiently learns homogeneous halfspaces in the presence of adversarial label noise with respect to a broad family of well-behaved distributions, including log-concave distributions. Before we state our contributions, we provide some background and motivation for this work.

A (homogeneous) halfspace is any function $f : \mathbb{R}^d \to \{\pm 1\}$ of the form $f(\mathbf{x}) = \mathrm{sign}(\langle \mathbf{w}, \mathbf{x} \rangle)$, where the vector $\mathbf{w} \in \mathbb{R}^d$ is called the weight vector of $f$, and the function $\mathrm{sign} : \mathbb{R} \to \{\pm 1\}$ is defined as $\mathrm{sign}(t) = 1$ if $t \geq 0$ and $\mathrm{sign}(t) = -1$ otherwise. Halfspaces are arguably the most fundamental concept class and have been studied since the beginning of machine learning, starting with the Perceptron algorithm [Ros58, Nov62]. In the realizable setting, halfspaces are efficiently learnable in the distribution-independent PAC model [Val84] via linear programming (see, e.g., [MT94]). On the other hand, in the agnostic model [Hau92, KSS94], even *weak* distribution-independent learning is computationally intractable [GR06, FGKP06, Dan16]. The distribution-specific agnostic (or adversarial label noise) setting – where the label noise is adversarial but we have some prior knowledge about the structure of the marginal distribution on examples – lies in between these two extremes. In this setting, computationally efficient noise-tolerant learning algorithms are known [KKMS08, KLS09a, ABL17, Dan15, DKS18] under various distributional assumptions. We start by defining the distribution-specific agnostic model.

**Definition 1.1** (Distribution-Specific PAC Learning with Adversarial Label Noise). Given i.i.d. labeled examples $(\mathbf{x}, y)$ from a distribution $\mathcal{D}$ on $\mathbb{R}^d \times \{\pm 1\}$, such that the marginal distribution $\mathcal{D}_\mathbf{x}$ is promised to belong in a known family $\mathcal{F}$ but the labels $y$ can be arbitrary, the goal of the learner

is to output a hypothesis $h$ with small misclassification error $\mathrm{err}_{0-1}^{\mathcal{D}}(h) \overset{\text{def}}{=} \mathbf{Pr}_{(\mathbf{x},y)\sim\mathcal{D}}[h(\mathbf{x}) \neq y]$, compared to $\mathrm{opt} \overset{\text{def}}{=} \inf_{g\in\mathcal{C}} \mathrm{err}_{0-1}^{\mathcal{D}}(g)$, where $\mathcal{C}$ is the target concept class.

[KKMS08] gave an algorithm that learns halfspaces in this model with error $\mathrm{opt} + \epsilon$ under any isotropic log-concave distribution, with sample complexity and runtime $d^{m(1/\epsilon)}$, for an appropriate function $m$, which is at least polynomial. Moreover, there is evidence that any algorithm that achieves error $\mathrm{opt} + \epsilon$ requires time *exponential* in $1/\epsilon$, even under Gaussian marginals [DKZ20, GGK20]. Specifically, recent work [DKZ20, GGK20] obtained Statistical Query (SQ) lower bounds of $d^{\mathrm{poly}(1/\epsilon)}$ for this problem.

A line of work [KLS09a, ABL17, Dan15, DKS18] focused on obtaining $\mathrm{poly}(d, 1/\epsilon)$ time algorithms with near-optimal error guarantees. Specifically, [ABL17] gave a polynomial time *constant-factor* approximation algorithm – i.e., an algorithm with misclassification error of $C \cdot \mathrm{opt} + \epsilon$, for some universal constant $C > 1$ – for homogeneous halfspaces under any isotropic log-concave distribution. More recent work [DKS18] gave an algorithm achieving this error bound for arbitrary halfspaces under Gaussian marginals. The algorithms of [ABL17, DKS18] rely on an iterative localization technique and are quite sophisticated. Moreover, while their complexity is polynomial, they do not appear to be practical. The motivation for this work is the design of simple and practical algorithms for this problem with the same near-optimal error guarantees as these prior works.

## 1.2 Our Contributions

Our main result is that SGD on a non-convex surrogate of the zero-one loss solves the problem of learning a homogeneous halfspace with adversarial label noise when the underlying marginal distribution on the examples is well-behaved. As we already mentioned, prior work [ABL17, DKS18] uses more complex methods and custom algorithms that run in multiple phases using multiple passes over the samples. In contrast, we take a direct optimization approach and define a single loss function over the space of halfspaces whose approximate stationary points are near-optimal solutions. This implies that *any optimization method* that is guaranteed to converge to stationary points, for example SGD, will yield a halfspace with error $O(\mathrm{opt}) + \epsilon$.

Our loss function is a smooth version of the 0-1 loss using a sigmoid function. In our case, we use the logistic function $S_\sigma(t) = 1/(1 + e^{-t/\sigma})$. Our overall objective is:

$$\mathcal{L}_\sigma(\mathbf{w}) = \underset{(\mathbf{x},y)\sim\mathcal{D}}{\mathbf{E}} \left[ S_\sigma\left(-y\left\langle\mathbf{w},\mathbf{x}\right\rangle\right) \right] , \tag{1}$$

and we optimize it over the unit sphere $\|\mathbf{w}\|_2 = 1$. We show that, for a broad class of distributions, any stationary point of this loss function corresponds to a halfspace with near-optimal error. In more detail, we require that the distribution on the examples is sufficiently well-behaved (Definition 1.2) satisfying natural (anti-)concentration properties.

In [DKTZ20], it was shown that the (approximate) stationary points of the objective of Equation (1) are (approximately) optimal halfspaces under Massart noise, which is a milder noise assumption than adversarial label noise. Interestingly, our results suggest that optimizing this objective is a unified approach for learning halfspaces under label noise, as we show that it works even in the more challenging adversarial noise setting.

**Definition 1.2** (Well-behaved distributions). Let $U, R > 0$ be absolute constants and $t : \mathbb{R}_+ \to \mathbb{R}_+$ be a non-negative function. An isotropic (i.e., zero mean and identity covariance) distribution $\mathcal{D}_{\mathbf{x}}$ on $\mathbb{R}^d$ is called well-behaved if for any projection $(\mathcal{D}_{\mathbf{x}})_V$ of $\mathcal{D}_{\mathbf{x}}$ onto a 2-dimensional subspace $V$ the corresponding pdf $\gamma_V$ on $\mathbb{R}^2$ satisfies the following properties:

1. $\gamma_V(\mathbf{x}) \geq 1/U$, for all $\mathbf{x} \in V$ such that $\|\mathbf{x}\|_2 \leq R$ (anti-anti-concentration).

2. For all $\mathbf{x} \in V$, we have $\gamma_V(\mathbf{x}) \leq t\left(\|\mathbf{x}\|_2\right)$ and also $\sup_{\mathbf{x}\in V} t(\|\mathbf{x}\|_2) \leq U$, $\int_V t(\|\mathbf{x}\|_2)\mathrm{d}\mathbf{x} \leq U$, $\int_V \|\mathbf{x}\|_2 t(\|\mathbf{x}\|_2)\mathrm{d}\mathbf{x} \leq U$ (anti-concentration and concentration).

Our class of distributions contains well-known distribution classes such as Gaussian and log-concave. In addition to distributions with strong concentration properties, our results also handle distributions with very weak concentration such as heavy-tailed distributions. In particular, we handle distributions whose density function decays only polynomially with the distance from the origin, see Table 1.

We use the *non-convex* objective of Equation (1) and SGD to obtain our main algorithmic result.

**Theorem 1.3.** *Let $\mathcal{D}$ be a distribution on $\mathbb{R}^d \times \{\pm 1\}$ such that the marginal $\mathcal{D}_\mathbf{x}$ on $\mathbb{R}^d$ is well-behaved. Then SGD on the objective* (1) *has the following performance guarantee: For any $\epsilon > 0$, it draws $m = \widetilde{O}(d/\epsilon^4)$ labeled examples from $\mathcal{D}$, uses $O(m)$ gradient evaluations, and outputs a hypothesis halfspace with misclassification error $O(\mathrm{opt}) + \epsilon$ with probability at least 99%.*

Theorem 1.3 gives a simple and practical learning algorithm for halfspaces with adversarial label noise with respect to a broad family of marginal distributions.

A natural question is whether the non-convexity of our surrogate loss (1) is required. In many practical settings, convex surrogates of the $0/1$ loss such as Hinge or ReLU loss are used, see [BJM06] for more choices. In general, given a convex and increasing loss $\ell(\cdot)$ the following objective is defined.

$$\mathcal{C}(\mathbf{w}) = \mathop{\mathbf{E}}_{(\mathbf{x},y)\sim\mathcal{D}}[\ell(-y\langle\mathbf{x},\mathbf{w}\rangle)]. \tag{2}$$

One such convex optimization problem closely related to our non-convex formulation is *logistic regression*. In that case, the convex surrogate is simply $\ell(t) = \log(S_\sigma(t))$ (compare with Equation (1)).

To complement our positive result, we show that convex surrogates are insufficient for the task at hand. *In particular, for any convex surrogate objective, one will obtain a halfspace with error $\omega(\mathrm{opt})$.* In more detail, we construct a single noisy distribution whose $\mathbf{x}$-marginal is well-behaved such that optimizing any convex objective over this distribution will yield a halfspace with error $\omega(\mathrm{opt})$. We establish a fine-grained result showing that the misclassification error of convex objectives degrades as the distributions become more *heavy tailed*, see Table 1.

**Theorem 1.4.** *Let $\mathcal{D}_\mathbf{x}$ be the standard normal distribution on $\mathbb{R}^d$. There exists a distribution $\mathcal{D}$ on $\mathbb{R}^d \times \{\pm 1\}$ such that for* every *convex and non-decreasing loss $\ell(\cdot)$ the objective $\mathcal{C}(\mathbf{w}) = \mathbf{E}_{(\mathbf{x},y)\sim\mathcal{D}}[\ell(-y\langle\mathbf{x},\mathbf{w}\rangle)]$ is minimized at some halfspace $h$ with misclassification error $\Omega(\mathrm{opt}\sqrt{\log(1/\mathrm{opt})})$. Moreover, if the marginal $\mathcal{D}_\mathbf{x}$ is allowed to be log-concave (resp. $s$-heavy tailed, $s > 2$) the error of any minimizer is $\Omega(\mathrm{opt}\log(1/\mathrm{opt}))$ (resp. $\Omega(\mathrm{opt}^{1-1/s})$).*

In fact, our lower bound result shows a strong statement about convex surrogates: Even under the nicest distribution possible, i.e., a Gaussian, there is some simple label noise (flipping the labels of points far from the origin) *that does not depend on the convex loss $\ell(\cdot)$* such that no convex objective can achieve $O(\mathrm{opt})$ error. This suggests that the shortcoming of convex objectives is not due to pathological cases and complicated noise distributions that are designed to fool each specific loss function, but is rather inherent.

Table 1: Common *well-behaved* distribution families with their corresponding parameters $U, R, t(\cdot)$, see Definition 1.2. The last two columns show the best possible error achievable by convex objectives and our non-convex objective of Eq.(1).

| **Distribution** | $U, R$ | $t(\mathbf{x})$ | Any Convex Loss | Our Loss, Eq.(1) |
|---|---|---|---|---|
| Gaussian | $\Theta(1)$ | $e^{-\Omega(\|\mathbf{x}\|_2^2)}$ | $\Omega(\mathrm{opt}\sqrt{\log(1/\mathrm{opt})})$ [Thm 1.4] | $O(\mathrm{opt})$ [Thm 1.3] |
| Log-Concave | $\Theta(1)$ | $e^{-\Omega(\|\mathbf{x}\|_2)}$ | $\Omega(\mathrm{opt}\log(1/\mathrm{opt}))$ [Thm 1.4] | $O(\mathrm{opt})$ [Thm 1.3] |
| $s$-Heavy Tailed, $s > 2$ | $\Theta(1)$ | $\frac{O(1)}{(\|\mathbf{x}\|_2+1)^{2+s}}$ | $\Omega(\mathrm{opt}^{1-1/s})$ [Thm 1.4] | $O(\mathrm{opt})$ [Thm 1.3] |

## 1.3 Overview of Techniques

Our approach is inspired by the recent work [DKTZ20], where the authors use the same loss function for learning halfspaces under the (weaker) Massart noise model. Under similar distributional assumptions to the ones we consider here, [DKTZ20] shows that the gradient of the loss function points towards the parameters of the optimal halfspace. A major difference between the two settings is that under Massart noise there exists a *unique optimal halfspace* and is identifiable. In the agnostic setting, there may be multiple halfspaces achieving optimal error. However, as we show, for the class of distributions we consider, all these solutions lie on a small cone, see Claim 3.4 establishing that the angle between any two halfspaces is small. Our algorithm aims to move towards the cone with every gradient step.

To achieve this, we must carefully set the parameter $\sigma$ of the objective. Smaller values of sigma amplify the contribution to the gradient of points closer to the current guess and enable using local information to obtain good gradients. This localization approach is necessary and is commonly used to efficiently learn halfspaces under structured distributions [ABL17, DKS18]. In the Massart model, the authors of [DKTZ20] show that for the loss function of Equation (1) any sufficiently small value for $\sigma$ suffices to obtain a gradient pointing towards the optimal halfspace. This is not true in the agnostic setting that we consider here. In particular, choosing small values of $\sigma$ may put a lot of weight on points close to the halfspace that may all be noisy. To prove our structural result, we show that there exists an appropriate setting of a not-too-small $\sigma$ that will guarantee convergence to a solution with $O(\mathrm{opt})$ error. This is our main structural result, Lemma 3.2.

Our lower bound hinges on the fact that such a trade-off can only be achieved using non-convex loss functions. In particular, our lower bound construction leverages the structure of convex objectives to design a noisy distribution where any convex objective results in misclassification error $\omega(\mathrm{opt})$. In more detail, we exploit the fact that all optimal halfspaces lie in a small cone, and show that there exists a fixed noise distribution such that all convex loss functions have non-zero gradients inside this cone.

## 1.4 Related Work

Here we provide a detailed summary of the most relevant prior work with a focus on $\mathrm{poly}(d/\epsilon)$ time algorithms. [KLS09b] studied the problem of learning homogeneous halfspaces in the adversarial label noise model, when the marginal distribution on the examples is isotropic log-concave, and gave a polynomial-time algorithm with error guarantee $\tilde{O}(\mathrm{opt}^{1/3}) + \epsilon$. This error bound was improved by [ABL17] who gave an efficient localization-based algorithm that learns to accuracy $O(\mathrm{opt}) + \epsilon$ for isotropic log-concave distributions. [DKS18] gave a localization-based algorithm that learns arbitrary halfspaces with error $O(\mathrm{opt}) + \epsilon$ for Gaussian marginals. [BZ17] extended the algorithms of [ABL17] to the class of $s$-concave distributions, for $s > -\Omega(1/d)$. Inspired by the localization approach, [YZ17] gave a perceptron-like learning algorithm that succeeds under the uniform distribution on the sphere. The algorithm of [YZ17] takes $\tilde{O}(d/\epsilon)$ samples, runs in time $\tilde{O}(d^2/\epsilon)$, and achieves error of $\tilde{O}(\log d \cdot \mathrm{opt}) + \epsilon$ – scaling logarithmically with the dimension $d$. We also note that [DKTZ20] established a structural result regarding the sufficiency of stationary points for learning homogeneous halfspaces with Massart noise. Finally, we draw an analogy with recent work [DGK+20] which established that convex surrogates suffice to obtain error $O(\mathrm{opt}) + \epsilon$ for the related problem of agnostically learning ReLUs under well-behaved distributions. This positive result for ReLUs stands in sharp contrast to the case of sign activations studied in this paper (as follows from our lower bound result). An interesting direction is to explore the effect of non-convexity for other common activation functions.

## 2 Preliminaries and Notation

For $n \in \mathbb{Z}_+$, let $[n] \stackrel{\mathrm{def}}{=} \{1, \ldots, n\}$. We will use small boldface characters for vectors. For $\mathbf{x} \in \mathbb{R}^d$ and $i \in [d]$, $\mathbf{x}_i$ denotes the $i$-th coordinate of $\mathbf{x}$, and $\|\mathbf{x}\|_2 \stackrel{\mathrm{def}}{=} (\sum_{i=1}^d \mathbf{x}_i^2)^{1/2}$ denotes the $\ell_2$-norm of $\mathbf{x}$. We will use $\langle \mathbf{x}, \mathbf{y} \rangle$ for the inner product of $\mathbf{x}, \mathbf{y} \in \mathbb{R}^d$ and $\theta(\mathbf{x}, \mathbf{y})$ for the angle between $\mathbf{x}, \mathbf{y}$. We will also denote $\mathbb{1}_A$ to be the characteristic function of the set $A$, i.e., $\mathbb{1}_A(\mathbf{x}) = 1$ if $\mathbf{x} \in A$ and $\mathbb{1}_A(\mathbf{x}) = 0$ if $\mathbf{x} \notin A$. Let $\mathbf{e}_i$ be the $i$-th standard basis vector in $\mathbb{R}^d$. Let $\mathrm{proj}_U(\mathbf{x})$ be the projection of $\mathbf{x}$ onto subspace $U \subset \mathbb{R}^d$. Let $\mathbf{E}[X]$ denote the expectation of random variable $X$ and $\mathbf{Pr}[\mathcal{E}]$ the probability of event $\mathcal{E}$. We consider the binary classification setting where labeled examples $(\mathbf{x}, y)$ are drawn i.i.d. from a distribution $\mathcal{D}$ on $\mathbb{R}^d \times \{\pm 1\}$. We denote by $\mathcal{D}_\mathbf{x}$ the marginal of $\mathcal{D}$ on $\mathbf{x}$. The misclassification error of a hypothesis $h : \mathbb{R}^d \to \{\pm 1\}$ (with respect to $\mathcal{D}$) is $\mathrm{err}_{0-1}^{\mathcal{D}}(h) \stackrel{\mathrm{def}}{=} \mathbf{Pr}_{(\mathbf{x},y) \sim \mathcal{D}}[h(\mathbf{x}) \neq y]$. The zero-one error between two functions $f, h$ (with respect to $\mathcal{D}_\mathbf{x}$) is $\mathrm{err}_{0-1}^{\mathcal{D}_\mathbf{x}}(f, h) \stackrel{\mathrm{def}}{=} \mathbf{Pr}_{\mathbf{x} \sim \mathcal{D}_\mathbf{x}}[f(\mathbf{x}) \neq h(\mathbf{x})]$.

## 3 Non-Convex SGD Learns Halfspaces with Adversarial Noise

In this section, we prove our main algorithmic result, whose formal version we restate here.

**Theorem 3.1.** *Let $\mathcal{D}$ be a distribution on $\mathbb{R}^d \times \{\pm 1\}$ such that the marginal $\mathcal{D}_{\mathbf{x}}$ on $\mathbb{R}^d$ is well-behaved. There is an algorithm with the following performance guarantee: For any $\epsilon > 0$, it draws $m = \widetilde{O}(d \log(1/\delta)/\epsilon^4)$ labeled examples from $\mathcal{D}$, uses $O(m)$ gradient evaluations, and outputs a hypothesis vector $\bar{\mathbf{w}}$ that satisfies $\mathrm{err}_{0-1}^{\mathcal{D}}(h_{\bar{\mathbf{w}}}) \leq O(\mathrm{opt}) + \epsilon$ with probability at least $1 - \delta$, where $\mathrm{opt}$ is the minimum classification error achieved by halfspaces.*

The crucial component in the proof of Theorem 3.1 is the following structural lemma, Lemma 3.2. We show that by carefully choosing the parameter $\sigma > 0$ of the non-convex surrogate loss $S_\sigma$ of Equation (1), we get that any approximate stationary point of this objective will be close to some optimal halfspace. Instead of optimizing over the unit sphere, we can normalize our objective $\mathcal{L}_\sigma$ defined in Equation (1), as follows

$$\mathcal{L}_\sigma(\mathbf{w}) = \underset{(\mathbf{x},y)\sim\mathcal{D}}{\mathbf{E}}\left[S_\sigma\left(-y\frac{\langle \mathbf{w}, \mathbf{x}\rangle}{\|\mathbf{w}\|_2}\right)\right], \tag{3}$$

where $S_\sigma(t) = \frac{1}{1+e^{-t/\sigma}}$ is the logistic function with growth rate $1/\sigma$. We prove the following:

**Lemma 3.2** (Stationary points of $\mathcal{L}_\sigma$ suffice). *Let $\mathcal{D}_{\mathbf{x}}$ be a well-behaved distribution on $\mathbb{R}^d$ and let $\mathbf{w}^*$ be a halfspace achieving optimal classification error $\mathrm{opt}$. Fix $\sigma > 0$ and let $\theta = (4\sqrt{2}\pi U/R) \cdot \sigma$. If $\mathrm{opt} \leq R^4/(2^{15}U^3) \cdot \sigma$, then for every $\widehat{\mathbf{w}}$ such that $\theta(\widehat{\mathbf{w}}, \mathbf{w}^*) \in (\theta, \pi - \theta)$ it holds $\|\nabla_{\mathbf{w}}\mathcal{L}_\sigma(\widehat{\mathbf{w}})\|_2 \geq \frac{R^2}{64U}$.*

*Proof Sketch.* To simplify notation, we will write $h(\mathbf{w}, \mathbf{x}) = \frac{\langle \mathbf{w}, \mathbf{x}\rangle}{\|\mathbf{w}\|_2}$. Note that $\nabla_{\mathbf{w}} h(\mathbf{w}, \mathbf{x}) = \frac{\mathbf{x}}{\|\mathbf{w}\|_2} - \langle \mathbf{w}, \mathbf{x}\rangle \frac{\mathbf{w}}{\|\mathbf{w}\|_2^3}$. We define the "noisy" region $S$, as follows $S = \{\mathbf{x} \in \mathbb{R}^d : y \neq \mathrm{sign}(\langle \mathbf{w}^*, \mathbf{x}\rangle)\}$. The gradient of the objective $\mathcal{L}_\sigma(\mathbf{w})$ is

$$\nabla_{\mathbf{w}}\mathcal{L}_\sigma(\mathbf{w}) = \underset{(\mathbf{x},y)\sim\mathcal{D}}{\mathbf{E}}\left[-S_\sigma'\left(-y\,h(\mathbf{w},\mathbf{x})\right)\nabla_{\mathbf{w}}h(\mathbf{w},\mathbf{x})\,y\right]$$

$$= \underset{\mathbf{x}\sim\mathcal{D}_{\mathbf{x}}}{\mathbf{E}}\left[-S_\sigma'\left(|h(\mathbf{w},\mathbf{x})|\right)\,\nabla_{\mathbf{w}}h(\mathbf{w},\mathbf{x})\,(1 - 2\cdot\mathbb{1}_S(\mathbf{x}))\,\mathrm{sign}(\langle\mathbf{w}^*,\mathbf{x}\rangle)\right].$$

Let $V = \mathrm{span}(\mathbf{w}^*, \mathbf{w})$. Since projections can only decrease the norm of a vector, we have $\|\nabla_{\mathbf{w}}\mathcal{L}_\sigma(\mathbf{w})\|_2 \geq \|\mathrm{proj}_V \nabla_{\mathbf{w}}\mathcal{L}_\sigma(\mathbf{w})\|_2$. Without loss of generality, we may assume that $\widehat{\mathbf{w}} = \mathbf{e}_2$ and $\mathbf{w}^* = -\sin\theta \cdot \mathbf{e}_1 + \cos\theta \cdot \mathbf{e}_2$. Then, we have $\mathrm{proj}_V(h(\mathbf{w},\mathbf{x})) = (\mathbf{x}_1, 0)$. Using the above and the triangle inequality, we obtain

$$\|\nabla_{\mathbf{w}}\mathcal{L}_\sigma(\mathbf{w})\|_2 \geq \underbrace{\left\|\underset{\mathbf{x}\sim\mathcal{D}_{\mathbf{x}}}{\mathbf{E}}\left[-S_\sigma'\left(|h(\mathbf{w},\mathbf{x})|\right)\,(\mathbf{x}_1,0)\,\mathrm{sign}(\langle\mathbf{w}^*,\mathbf{x}\rangle)\right]\right\|_2}_{I_1}$$

$$- 2\underbrace{\left\|\underset{\mathbf{x}\sim\mathcal{D}_{\mathbf{x}}}{\mathbf{E}}\left[-\mathbb{1}_S(\mathbf{x})S_\sigma'\left(|h(\mathbf{w},\mathbf{x})|\right)\,(\mathbf{x}_1,0)\,\mathrm{sign}(\langle\mathbf{w}^*,\mathbf{x}\rangle)\right]\right\|_2}_{I_2}.$$

Let $R, U$ be absolute constants from the Definition 1.2. We will first bound from above the term $I_2$, i.e., the contribution of the noisy points to the gradient. Using the fact that $S_\sigma'(|t|) \leq e^{-|t|/\sigma}/\sigma$ we obtain

$$I_2 \leq \underset{\mathbf{x}\sim\mathcal{D}_{\mathbf{x}}}{\mathbf{E}}\left[\frac{e^{-|\mathbf{x}_2|/\sigma}}{\sigma}|\mathbf{x}_1|\,\mathbb{1}_S(\mathbf{x})\right] \leq \sqrt{\underset{\mathbf{x}\sim\mathcal{D}_{\mathbf{x}}}{\mathbf{E}}[\mathbb{1}_S(\mathbf{x})]}\sqrt{\underset{\mathbf{x}\sim\mathcal{D}_{\mathbf{x}}}{\mathbf{E}}\left[\frac{e^{-2|\mathbf{x}_2|/\sigma}}{\sigma^2}\mathbf{x}_1^2\right]}$$

$$\leq \sqrt{\frac{\mathrm{opt}}{\sigma}}\sqrt{\underset{\mathbf{x}\sim(\mathcal{D}_{\mathbf{x}})_V}{\mathbf{E}}\left[\frac{e^{-2|\mathbf{x}_2|/\sigma}}{\sigma}\mathbf{x}_1^2\right]}.$$

where the first inequality follows from the Cauchy-Schwarz inequality and for the second we used the fact that the set $S$ has probability at most $\mathrm{opt}$. To finish the bound, notice that the remaining expectation depends only on $\mathbf{x}_1, \mathbf{x}_2$ and therefore we can use the upper bound $t(\cdot)$ on the density

function. Using polar coordinates we obtain

$$\mathop{\mathbf{E}}_{\mathbf{x}\sim(\mathcal{D}_{\mathbf{x}})_V}\left[\frac{e^{-2|\mathbf{x}_2|/\sigma}}{\sigma}\mathbf{x}_1^2\right] \leq 4\int_0^\infty \int_0^{\pi/2} \frac{r^3}{\sigma}\cos^2(\phi)e^{-2r\sin(\phi)/\sigma}t(r)\mathrm{d}\phi\mathrm{d}r$$

$$\leq 2\int_0^\infty r^2 t(r)\int_0^{\pi/2}\frac{2r}{\sigma}\cos(\phi)e^{-2r\sin(\phi)/\sigma}\mathrm{d}\phi\mathrm{d}r$$

$$= 2\int_0^\infty r^2 t(r)(1-e^{-2r/\sigma})\mathrm{d}r \leq 2\int_0^\infty r^2\, t(r)\mathrm{d}r \leq 2U\,,$$

where for the last inequality we used the fact that $1 - e^{-2r/\sigma} \leq 1$. We thus have $I_2 \leq \sqrt{2U\mathrm{opt}/\sigma}$.
We now bound $I_1$ from below. Observe that since inner products with $\mathbf{w}^*$, $\mathbf{w}$ are preserved when we project $\mathbf{x}$ to $V$, we have $I_1 = \left|\mathbf{E}_{\mathbf{x}\sim(\mathcal{D}_{\mathbf{x}})_V}[S'_\sigma(|\mathbf{x}_2|)\mathbf{x}_1\mathrm{sign}(\langle\mathbf{w}^*,\mathbf{x}\rangle)]\right|$. Now, if we define $G = \{(\mathbf{x}_1,\mathbf{x}_2)\in\mathbb{R}^2 : \mathbf{x}_1\mathrm{sign}(\langle\mathbf{w}^*,\mathbf{x}\rangle) > 0\}$, using the triangle inequality we have

$$I_1 \geq \mathop{\mathbf{E}}_{\mathbf{x}\sim(\mathcal{D}_{\mathbf{x}})_V}[S'_\sigma(|\mathbf{x}_2|)|\mathbf{x}_1|\mathbb{1}_G(\mathbf{x})] - \mathop{\mathbf{E}}_{\mathbf{x}\sim(\mathcal{D}_{\mathbf{x}})_V}[S'_\sigma(|\mathbf{x}_2|)|\mathbf{x}_1|\mathbb{1}_{G^c}(\mathbf{x})]\,.$$

Moreover, using the fact that $e^{-|t|/\sigma}/(4\sigma) \geq S'_\sigma(|t|) \leq e^{-|t|/\sigma}/\sigma$ we get

$$I_1 \geq \frac{1}{4}\mathop{\mathbf{E}}_{\mathbf{x}\sim(\mathcal{D}_{\mathbf{x}})_V}\left[|\mathbf{x}_1|\mathbb{1}_G(\mathbf{x})e^{-|\mathbf{x}_2|/\sigma}/\sigma\right] - \mathop{\mathbf{E}}_{\mathbf{x}\sim(\mathcal{D}_{\mathbf{x}})_V}\left[|\mathbf{x}_1|\mathbb{1}_{G^c}(\mathbf{x})e^{-|\mathbf{x}_2|/\sigma}/\sigma\right]\,. \qquad (4)$$

We can now bound each term separately using the fact that the distribution $\mathcal{D}_{\mathbf{x}}$ is well-behaved. Similarly to the previous bound, one can use polar coordinates to obtain

$$\mathop{\mathbf{E}}_{\mathbf{x}\sim(\mathcal{D}_{\mathbf{x}})_V}\left[e^{-|\mathbf{x}_2|/\sigma}|\mathbf{x}_1|\,\mathbb{1}_G(\mathbf{x})/\sigma\right] \geq \frac{R^2}{4U} \quad\text{and}\quad \mathop{\mathbf{E}}_{\mathbf{x}\sim(\mathcal{D}_{\mathbf{x}})_V}\left[\frac{e^{-|\mathbf{x}_2|/\sigma}}{\sigma}|\mathbf{x}_1|\,\mathbb{1}_{G^c}(\mathbf{x})\right] \leq \frac{2U\sigma^2}{\sin^2\theta}\,.$$

Putting everything in (4), we obtain $I_1 \geq R^2/(16U) - 2U\sigma^2/\sin^2\theta$. Notice now that the case where $\theta(\widehat{\mathbf{w}},\mathbf{w}^*) \in (\pi/2, \pi-\theta)$ follows similarly. Finally, in the case where $\theta = \pi/2$, the region $G^c$ is empty, and we again get the same lower bound on the gradient. Let $A > 0$, and set $\theta = A\cdot\sigma < \pi/2$, and let $\tau = \mathrm{opt}/\sigma$. Since $\sin(t) \geq 2t/\pi$ for every $t\in[0,\pi/2]$, we have $I_1 - 2I_2 \geq \frac{R^2}{16U} - \frac{\pi^2 U}{2A^2} - 2\sqrt{2U\tau}$.

For $\tau \leq \frac{R^4}{2^{15}U^3}$ and $A \geq 4\sqrt{2}\pi U/R$, it holds $I_1 - 2I_2 \geq R^2/(32U)$. $\qquad\square$

Using Lemma 3.2 we get our main algorithmic result. Our algorithm proceeds by Projected Stochastic Gradient Descent (PSGD), with projection on the $\ell_2$-unit sphere, to find an approximate stationary point of our non-convex surrogate loss. Since $\mathcal{L}_\sigma(\mathbf{w})$ is non-smooth for vectors $\mathbf{w}$ close to $\mathbf{0}$, at each step, we project the update on the unit sphere to avoid the region where the smoothness parameter is high. We are going to use the following result about the convergence of non-convex, smooth SGD on the unit sphere.

**Lemma 3.3** (Lemma 4.2 and 4.3 of [DKTZ20]). *Let $\mathcal{L}_\sigma(\mathbf{w})$ be as in Equation (1). After $T$ iterations, where $T = \Theta(d\log(1/\delta)/(\sigma^4\rho^4))$, the output $(\mathbf{w}^{(1)},\ldots,\mathbf{w}^{(T)})$ of Algorithm 1 satisfies $\min_{i=1,\ldots,T}\left\|\nabla_{\mathbf{w}}\mathcal{L}_\sigma(\mathbf{w}^{(i)})\right\|_2 \leq \rho$, with probability at least $1-\delta$.*

---

**Algorithm 1** PSGD for $f(\mathbf{w}) = \mathbf{E}_{\mathbf{z}\sim\mathcal{D}}[g(\mathbf{z},\mathbf{w})]$

---

1: **procedure** PSGD$(f, T, \beta)$ $\qquad\triangleright f(\mathbf{w}) = \mathbf{E}_{\mathbf{z}\sim\mathcal{D}}[g(\mathbf{z},\mathbf{w})]$: loss, $T$: number of steps, $\beta$: step size.
2: $\qquad \mathbf{w}^{(0)} \leftarrow \mathbf{e}_1$
3: $\qquad$ **for** $i = 1,\ldots,T$ **do**
4: $\qquad\qquad$ Sample $\mathbf{z}^{(i)}$ from $\mathcal{D}$.
5: $\qquad\qquad \mathbf{v}^{(i)} \leftarrow \mathbf{w}^{(i-1)} - \beta\nabla_{\mathbf{w}}g(\mathbf{z}^{(i)},\mathbf{w}^{(i-1)})$
6: $\qquad\qquad \mathbf{w}^{(i)} \leftarrow \mathbf{v}^{(i)}/\left\|\mathbf{v}^{(i)}\right\|_2$
7: $\qquad$ **return** $(\mathbf{w}^{(1)},\ldots,\mathbf{w}^{(T)})$.

---

In order to relate the miss-classification error of a candidate halpsace with the angle that it forms with an optimal halfpsace we are going to use the following claim that states that the disagreement error between two halfspaces is $\Theta(\theta(\mathbf{u},\mathbf{v}))$ under well-behaved distributions. Its proof can be found in the supplementary material.

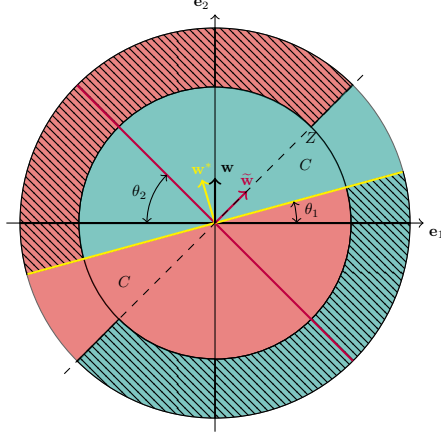

Figure 1: The green region depicts all points with $+1$ label and the red region depicts points with $-1$ label. We have $y = -\text{sign}(\langle \mathbf{w}^*, \mathbf{x} \rangle)$ for all points in $S \setminus C$, this corresponds to the hatched region. We have $\theta(\mathbf{w}^*, \mathbf{w}) = \theta_1$ and $\theta(\widetilde{\mathbf{w}}, \mathbf{w}) = \theta_2$.

**Claim 3.4.** *Let $\mathcal{D}_\mathbf{x}$ be a distribution on $\mathbb{R}^d$. Let $f \in \text{argmin}_{g \in \mathcal{C}} \text{err}_{0-1}^{\mathcal{D}}(g)$, where $\mathcal{C}$ is the class of halfspaces, then for any $\mathbf{u} \in \mathbb{R}^d$, it holds that $\text{err}_{0-1}^{\mathcal{D}_\mathbf{x}}(h_\mathbf{u}, f) - \text{err}_{0-1}^{\mathcal{D}}(f) \leq \text{err}_{0-1}^{\mathcal{D}}(h_\mathbf{u}) \leq \text{err}_{0-1}^{\mathcal{D}}(f) + \text{err}_{0-1}^{\mathcal{D}_\mathbf{x}}(h_\mathbf{u}, f)$. Moreover, if the distribution $\mathcal{D}_\mathbf{x}$ is well-behaved, then $\text{err}_{0-1}^{\mathcal{D}_\mathbf{x}}(h_\mathbf{u}, h_\mathbf{v}) = \Theta(\theta(\mathbf{u}, \mathbf{v}))$.*

Now assuming that we know the value of opt, we can readily use SGD and obtain a halfspace with small classification error. The following lemma, which relies on Claim 3.4, shows that SGD will output a list of candidate vectors, one of which will have error $\text{opt} + O(\sigma)$. For our structural result to work, we need $\text{opt} \leq C\sigma$ which gives the $O(\text{opt})$ error overall. Recall that for all well-behaved distributions the parameters $U, R$ are absolute constants. Its proof can be found in the Supplementary Material.

**Lemma 3.5.** *Let $\mathcal{D}$ be a distribution on $\mathbb{R}^d \times \{\pm 1\}$ such that the marginal $\mathcal{D}_\mathbf{x}$ on $\mathbb{R}^d$ is well-behaved. Algorithm 1 has the following performance guarantee: If $\text{opt} \leq C \cdot \sigma$ where $C = \frac{R^4}{2^{15}U^3}$, it draws $m = \text{poly}(U/R) \cdot d\frac{\log(1/\delta)}{\sigma^4}$ labeled examples from $\mathcal{D}$, uses $O(m)$ gradient evaluations, and outputs a hypothesis list of vectors $L$, such that there exist a vector $\bar{\mathbf{w}} \in L$ that satisfies $\text{err}_{0-1}^{\mathcal{D}}(h_{\bar{\mathbf{w}}}) \leq \text{opt} + O(\sigma)$ with probability at least $1 - \delta$, where $\text{opt}$ is the minimum classification error achieved by halfspaces.*

We now sketch the proof of our main theorem, Theorem 1.3. For the full proof see Supplementary Material.

*Proof Sketch of Theorem 1.3.* Since we do not know the value of opt, we can make an $\epsilon$ grid of $[0, 1]$ and run the SGD, Algorithm 1. Then we can test the empirical classification error all candidates produced and pick the best one. This would result in an additional $1/\epsilon$ factor in the sample complexity. A simple way to avoid that is by doing binary search for $\sigma$ instead, which only increases sample complexity by a $\log(1/\epsilon)$ factor. □

## 4 Convex Objectives Do Not Work

In this section we show that optimizing convex surrogates of the zero-one loss cannot get error $O(\text{opt}) + \epsilon$. We first recall the agnostic PAC learning setting that we assume here. Given a distribution $\mathcal{D}_\mathbf{x}$ on $\mathbb{R}^d$ and a halfspace $\mathbf{w}^*$, we can define a noiseless instance $\mathcal{D}$ on $\mathbb{R}^d \times \{\pm 1\}$ by setting the label of each point $\mathbf{x}$ to $y = \text{sign}(\langle \mathbf{w}^*, \mathbf{x} \rangle)$. In this setting, $\mathbf{w}^*$ achieves 0 classification error. To get a distribution where $\mathbf{w}^*$ achieves error $\text{opt} > 0$, we can simply flip the labels of an opt fraction of points $\mathbf{x}$. We now restate our theorem and give a sketch of its proof. The full proof can be found in the supplementary material.

**Theorem 4.1.** *Let $\mathcal{D}_\mathbf{x}$ be a standard Normal distribution on $\mathbb{R}^d$. There exist a distribution $\mathcal{D}$ on $\mathbb{R}^d \times \{\pm 1\}$ such that for every convex, non-decreasing, loss $\ell(\cdot)$ the objective $\mathcal{C}(\mathbf{w}) = \mathbf{E}_{\mathbf{x},y \sim \mathcal{D}}[\ell(-y \langle \mathbf{x}, \mathbf{w}\rangle)]$, is minimized at some halfspace $h$ with error $\mathrm{err}_{0-1}^{\mathcal{D}}(h) = \Omega(\mathrm{opt}\sqrt{\log(1/\mathrm{opt})})$. Moreover, if the marginal $\mathcal{D}_\mathbf{x}$ is allowed to be log-concave (resp. s-heavy tailed, $s > 2$) the error of any minimizer is $\Omega(\mathrm{opt}\log(1/\mathrm{opt}))$ (resp. $\Omega(\mathrm{opt}^{1-1/s})$).*

*Proof Sketch.* Since all the examples that we are going to consider will be radially invariant distribution, we note that the "disagreement" error of two halfspaces with normal vectors $\mathbf{v}, \mathbf{u}$ is $\mathrm{err}_{0-1}^{\mathcal{D}_\mathbf{x}}(\mathbf{v}, \mathbf{u}) = \theta(\mathbf{v}, \mathbf{u})/\pi$. From Claim 3.4, we have that the classification error of any candidate $\mathbf{w}$ is lower bounded by $\theta(\mathbf{w}, \mathbf{w}^*)/\pi - \mathrm{opt}$. We will construct a distribution $\mathcal{D}$ such that there is some $\mathbf{w}^*$ that achieves error $\mathrm{opt}$ but at the same time $\mathcal{C}(\mathbf{w})$ is minimized at some halfspace that is in angle $\theta(\mathbf{w}, \mathbf{w}^*) = \omega(\mathrm{opt})$. This means that the minimizer of $\mathcal{C}$ has classification error $\omega(\mathrm{opt})$.

For this sketch, we assume that $\mathcal{D}_\mathbf{x}$ is a standard Normal and without loss of generality work in 2 dimensions, for the other cases see the Supplementary Material. It's density function is radially invariant, i.e. $\gamma(\mathbf{x}_1, \mathbf{x}_2) = \frac{1}{2\pi} e^{-\|\mathbf{x}\|_2^2/2}$. If $\ell$ is a constant function any halfspace would minimize it and therefore, this case is trivial.

Take any $\mathbf{w}$ such that $\theta_1 = \theta(\mathbf{w}, \mathbf{w}^*) \leq \theta$. We are going to lower bound the norm of the gradient of $\mathcal{C}$ at $\mathbf{w}$. The gradient of $\mathcal{C}(\mathbf{w})$ is $\nabla_\mathbf{w}\mathcal{C}(\mathbf{w}) = \mathbf{E}_{(\mathbf{x},y)\sim\mathcal{D}}[-y\mathbf{x}\,\ell'(-y\langle\mathbf{x},\mathbf{w}\rangle)]$. We start by constructing the noisy distribution $\mathcal{D}$. Fix any unit vector $\mathbf{w}^*$ and let $\widetilde{\mathbf{w}}$ be a vector such that $\theta(\mathbf{w}^*, \widetilde{\mathbf{w}}) = \theta_2$, where $2\theta \leq \theta_2 \leq \pi/4$. Denote by $\widetilde{\mathbf{w}}^\perp$ the vector that is perpendicular with $\widetilde{\mathbf{w}}$ and satisfies $\langle \mathbf{w}^*, \widetilde{\mathbf{w}}^\perp \rangle \geq 0$. We now define the region $C, S$ that will help us define the parts of the distribution where we will introduce noise by flipping the $y$-labels, see also Figure 1.

$$C = \left\{ \mathbf{x}: \ \langle \mathbf{w}^*, \mathbf{x}\rangle \langle \widetilde{\mathbf{w}}, \mathbf{x}\rangle \geq 0 \text{ and } \langle \widetilde{\mathbf{w}}^\perp, \mathbf{x}\rangle \leq 0 \right\} \qquad S = \left\{ \mathbf{x}: \ \|\mathbf{x}\|_2 \geq Z \right\}.$$

We are now ready to define our noisy distribution $\mathcal{D}$: *we flip the labels of all points in the set $S \setminus C$.* Observe that $\mathrm{err}_{0-1}^{\mathcal{D}}(\mathbf{w}^*) \leq \mathbf{Pr}_{\mathbf{x}\sim\mathcal{D}_\mathbf{x}}[\|\mathbf{x}\|_2 \geq Z]$. This means that for the Gaussian we need to pick $Z = \sqrt{2\log(1/\mathrm{opt})}$ to have optimal error less than opt. Take any $\mathbf{w}$ such that $\theta_1 = \theta(\mathbf{w}, \mathbf{w}^*) \leq \theta$. To simplify notation, we assume that the length of $\mathbf{w}$ is 1 and $\mathbf{w} = \mathbf{e}_2$. We have that the first coordinate of the gradient is

$$\langle \nabla_\mathbf{w}(\mathcal{C}(\mathbf{w}), \mathbf{e}_1\rangle = \mathop{\mathbf{E}}_{(\mathbf{x},y)\sim\mathcal{D}}[-y\mathbf{x}_1\,\ell'(-y\,\mathbf{x}_2)]. \tag{5}$$

We consider here the more interesting case where $\mathbf{w}$ lies between $\mathbf{w}^*$ and $\widetilde{\mathbf{w}}$ as shown in Figure 1. We first compute the contribution to the gradient in $S^c$, i.e., the points where $y = \mathrm{sign}(\langle \mathbf{w}^*, \mathbf{x}\rangle)$. Since the distribution is radially symmetric, one can use polar coordinates to compute

$$I_{S^c} = \mathop{\mathbf{E}}_{(\mathbf{x},y)\sim\mathcal{D}}[-y\mathbf{x}_1\,\ell'(-y\mathbf{x}_2)\mathbb{1}_{S^c}(\mathbf{x})] = 2\int_0^Z r\gamma(r)(\ell(r\,\sin\theta_1) - \ell(-r\,\sin\theta_1))\mathrm{d}r. \tag{6}$$

Observe that since $\ell(\cdot)$ is non-decreasing we have $I_{S^c} \geq 0$. Next we compute the contribution of region $S$ to the gradient. Recall, that $S$ contains $S \setminus C$, i.e. the region we flipped the labels, $y = -\mathrm{sign}(\langle \mathbf{w}^*, \mathbf{x}\rangle)$, see Figure 1. We have

$$I_S = \mathop{\mathbf{E}}_{(\mathbf{x},y)\sim\mathcal{D}}[-y\mathbf{x}_1\,\ell'(-y\mathbf{x}_2)\mathbb{1}_S(\mathbf{x})] = 2\int_Z^\infty r\gamma(r)(\ell(-r\,\cos\theta_2) - \ell(r\,\cos\theta_2)\mathrm{d}r. \tag{7}$$

Similarly, to the previous case the fact that $\ell(\cdot)$ is non-decreasing implies that $I_S \leq 0$.

Now, we are going to crucially use the convexity of $\ell(\cdot)$. Since both $\theta_1, \theta_2 \leq \pi/4$, we have that $\cos\theta_2 \geq \sin\theta_1$ and therefore, from convexity of $\ell(\cdot)$, we obtain

$$\frac{\ell(r\sin\theta_1) - \ell(-r\sin\theta_1)}{2r\sin\theta_1} \leq \frac{\ell(r\cos\theta_2) - \ell(-r\sin\theta_1)}{r\cos\theta_2 + r\sin\theta_1}.$$

Since $\ell(\cdot)$ is also non-decreasing, we have that $\ell(r\cos\theta_2) - \ell(-r\sin\theta_1) \leq \ell(r\cos\theta_2) - \ell(-r\cos\theta_2)$ and therefore,

$$\ell(r\sin\theta_1) - \ell(-r\sin\theta_1) \leq \frac{2\sin\theta_1}{\cos\theta_2 + \sin\theta_1}(\ell(r\cos\theta_2) - \ell(-r\cos\theta_2)).$$

To simplify notation, we define the functions $\bar{\ell}(r) = \ell(r\cos\theta_2)$ and $h(r) = \bar{\ell}(r) - \bar{\ell}(-r)$. Observe that $\bar{\ell}(\cdot)$ enjoys exactly the same properties as $\ell(\cdot)$, that is $\bar{\ell}(\cdot)$ is convex, non-decreasing, and non-constant. Moreover, note that $h(r)$ is non-negative and non-decreasing. Using Inequalities (6), (7), the fact that $\theta_1 \leq \theta$ and setting $\theta_2 = 2\theta$, we obtain that

$$\langle \nabla_{\mathbf{w}} \mathcal{C}(\mathbf{w}), \mathbf{e}_1 \rangle = I_S + I_{S^c} \leq O(\theta) \underbrace{\int_0^Z r\gamma(r)h(r)\mathrm{d}r}_{I_1} - \underbrace{\int_Z^\infty r\gamma(r)h(r)\mathrm{d}r}_{I_2} \ . \tag{8}$$

Recall now that $Z = \sqrt{2\log(1/\mathrm{opt})}$. If for that $Z$, we can show that $I_2/I_1 = \Omega(\mathrm{opt}\sqrt{\log(1/\mathrm{opt})})$, then we can pick $\theta = \Omega(I_2/I_1)$ and have that the gradient of any halfspace $\mathbf{w}$ with angle smaller than $\theta$ with $\mathbf{w}^*$ is non-zero and therefore these vectors *are not minimizers of* $\mathcal{C}$. To finish the sketch, we observe that the worst case convex and non-decreasing function $\ell(\cdot)$ so that the ratio $I_2/I_1$ is minimized is simply a shifted ReLU function, i.e. $\bar{\ell}(r) = \alpha\max(r - r_0) + \beta$ for some constants $\alpha, \beta, r_0$. For this $\bar{\ell}(\cdot)$ the ratio is indeed $\Omega(\mathrm{opt}\sqrt{\log(1/\mathrm{opt})})$ under the Gaussian density. $\qquad\square$

## Broader Impact

Our work fits within the broader agenda of designing efficient robust learners in the presence of noisy data. The practical significance of learning linear classifiers is well-established. In most use-cases, the labels of the examples are noisy, e.g., a non-spam email is incorrectly labeled as spam and vice versa. Machine learning algorithms are now used more than ever having applications in social networks, auctions, advertising, etc. Therefore, having erroneous results due to noise can have far reaching consequences. The goal of our work is to develop robust machine learning algorithms that have theoretical performance guarantees and, at the same time, are easy to implement and use in practice.

Since the primary focus of our work is theoretical, we do not expect our results to have immediate societal impact. Nonetheless, we believe that our algorithm is practical and provides interesting insights that could be useful in practice. We show that one can use a simple black-box optimization routine (Stochastic Gradient Descent) on a simple objective function and learn an accurate linear classifier, even if a small constant fraction of the labels is chosen adversarially. Moreover, while many of the linear classification methods currently used in practice rely on optimizing convex surrogate objectives, our work shows that such methods may achieve significantly sub-optimal results, even under very benign distributions. Our method outperforms the error guarantee of any method relying on optimizing convex losses.

## Acknowledgments and Disclosure of Funding

Ilias Diakonikolas is supported by NSF Award CCF-1652862 (CAREER), a Sloan Research Fellowship, and a DARPA Learning with Less Labels (LwLL) grant. Nikos Zarifis is supported in part by a DARPA Learning with Less Labels (LwLL) grant.

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
