[Supplementary Material]

**Supplementary Material**

 **A    Omitted Proofs from Section 3**

366    **A.1    Proof of Lemma 3.2**

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

387    Assume first that $\theta(\mathbf{w}^*, \widehat{\mathbf{w}}) = \theta \in (0, \pi/2)$. Then we can express the region $G$ in polar coordinates
388    as $G = \{(r, \phi) : \phi \in (0, \theta) \cup (\pi/2, \pi + \theta) \cup (3\pi/2, 2\pi)\}$.

389    We denote by $\gamma(x, y)$ the density of the 2-dimensional projection on $V$ of the marginal distribution
390    $\mathcal{D}_\mathbf{x}$. Since the integral is non-negative, we can bound from below the contribution of region $G$ on the
391    gradient by integrating over $\phi \in (\pi/2, \pi)$. Specifically, we have:

$$\mathop{\mathbf{E}}_{\mathbf{x} \sim (\mathcal{D}_\mathbf{x})_V} \left[\frac{e^{-|\mathbf{x}_2|/\sigma}}{\sigma} |\mathbf{x}_1| \mathbb{1}_G(\mathbf{x})\right] \geq \int_0^\infty \int_{\pi/2}^\pi \gamma(r\cos\phi, r\sin\phi) r^2 |\cos\phi| \frac{e^{-\frac{r\sin\phi}{\sigma}}}{\sigma} \mathrm{d}\phi \mathrm{d}r$$

$$= \int_0^\infty \int_0^{\pi/2} \gamma(r\cos\phi, r\sin\phi) r^2 \cos\phi \frac{e^{-\frac{r\sin\phi}{\sigma}}}{\sigma} \mathrm{d}\phi \mathrm{d}r$$

$$\geq \frac{1}{U} \int_0^R r^2 \mathrm{d}r \int_0^{\pi/2} \cos\phi \frac{e^{-\frac{R\sin\phi}{\sigma}}}{\sigma} \mathrm{d}\phi$$

$$= \frac{1}{3U} R^2 \left(1 - e^{-\frac{R}{\sigma}}\right) \geq \frac{1}{4U} R^2 , \tag{10}$$

392    where for the second inequality we used the lower bound $1/U$ on the density function $\gamma(x, y)$ (see
393    Definition 1.2) and for the last inequality we used that $\sigma \leq \frac{R}{8}$ and that $1 - e^{-8} \geq 3/4$.

394    We next bound from above the contribution of the gradient in region $G^c$. Note that $G^c = \{(r, \phi) :$
395    $\phi \in B_\theta = (\pi/2 - \theta, \pi/2) \cup (3\pi/2 - \theta, 3\pi/2)\}$. Hence, we can write:

$$\mathop{\mathbf{E}}_{\mathbf{x} \sim (\mathcal{D}_\mathbf{x})_V} \left[\frac{e^{-|\mathbf{x}_2|/\sigma}}{\sigma} |\mathbf{x}_1| \mathbb{1}_{G^c}(\mathbf{x})\right] = \frac{1}{\sigma} \int_0^\infty \int_{\phi \in B_\theta} \gamma(r\cos\phi, r\sin\phi) r^2 \cos\phi e^{-\frac{r\sin\phi}{\sigma}} \mathrm{d}\phi \mathrm{d}r$$

$$\leq \frac{2U}{\sigma} \int_0^\infty \int_\theta^{\pi/2} r^2 \cos\phi e^{-\frac{r\sin\phi}{\sigma}} \mathrm{d}\phi \mathrm{d}r$$

$$= \frac{2U\sigma^2 \cos^2\theta}{\sin^2\theta} , \tag{11}$$

396    where the inequality follows from the upper bound $U$ on the density $\gamma(x, y)$ (see Definition 1.2).
397    Putting everything in (9), we obtain $I_1 \geq R^2/(16U) - 2U\sigma^2/\sin^2\theta$. Notice now that the case where
398    $\theta(\widehat{\mathbf{w}}, \mathbf{w}^*) \in (\pi/2, \pi - \theta)$ follows similarly. Finally, in the case where $\theta = \pi/2$, the region $G^c$ is
399    empty, and we again get the same lower bound on the gradient. Let $A > 0$, and set $\theta = A \cdot \sigma < \pi/2$,
400    and let $\tau = \mathrm{opt}/\sigma$. Since $\sin(t) \geq 2t/\pi$ for every $t \in [0, \pi/2]$, we have

$$I_1 - 2I_2 \geq \frac{R^2}{16U} - \frac{\pi^2 U}{2A^2} - 2\sqrt{2U\tau} .$$

401    For $\tau \leq \frac{R^4}{2^{15}U^3}$ and $A \geq 4\sqrt{2}\pi U/R$, it holds $I_1 - 2I_2 \geq R^2/(32U)$.     $\square$

## A.2   Proof of Claim 3.4

403    *Proof.* Let $S = \{\mathbf{x} \in \mathbb{R}^d : y \neq f(\mathbf{x})\}$, then we have

$$\mathrm{err}_{0-1}^{\mathcal{D}_\mathbf{x}}(h_\mathbf{u}, f) = \int_{S^c} \mathbb{1}\{h_\mathbf{u}(\mathbf{x}) \neq y\} \gamma(\mathbf{x}) \mathrm{d}\mathbf{x} + \int_S \mathbb{1}\{h_\mathbf{u}(\mathbf{x}) = y\} \gamma(\mathbf{x}) \mathrm{d}\mathbf{x}$$

$$= \int_{\mathbb{R}^d} \mathbb{1}\{h_\mathbf{u}(\mathbf{x}) \neq y\} \gamma(\mathbf{x}) \mathrm{d}\mathbf{x} + 2 \int_S \mathbb{1}\{h_\mathbf{u}(\mathbf{x}) = y\} \gamma(\mathbf{x}) \mathrm{d}\mathbf{x} - \int_S \gamma(\mathbf{x}) \mathrm{d}\mathbf{x}$$

$$= \mathrm{err}_{0-1}^{\mathcal{D}}(h_\mathbf{u}) + 2 \int_S \mathbb{1}\{h_\mathbf{u}(\mathbf{x}) = y\} \gamma(\mathbf{x}) \mathrm{d}\mathbf{x} - \mathrm{err}_{0-1}^{\mathcal{D}}(f) .$$

Using that $\int_S \mathbb{1}\{h_{\mathbf{u}}(\mathbf{x}) = y\}\gamma(\mathbf{x})\mathrm{d}\mathbf{x} \geq 0$, the result follows. To prove that $\mathrm{err}_{0-1}^{\mathcal{D}_{\mathbf{x}}}(h_{\mathbf{u}}, f) - \mathrm{err}_{0-1}^{\mathcal{D}}(f) \leq \mathrm{err}_{0-1}^{\mathcal{D}}(h_{\mathbf{u}})$, we work as follows

$$
\begin{aligned}
\mathrm{err}_{0-1}^{\mathcal{D}_{\mathbf{x}}}(h_{\mathbf{u}}, f) &= \int_{S^c} \mathbb{1}\{h_{\mathbf{u}}(\mathbf{x}) \neq y\}\gamma(\mathbf{x})\mathrm{d}\mathbf{x} + \int_S \mathbb{1}\{h_{\mathbf{u}}(\mathbf{x}) = y\}\gamma(\mathbf{x})\mathrm{d}\mathbf{x} \\
&= \int_{\mathbb{R}^d} \mathbb{1}\{h_{\mathbf{u}}(\mathbf{x}) \neq y\}\gamma(\mathbf{x})\mathrm{d}\mathbf{x} + \int_S \gamma(\mathbf{x})\mathrm{d}\mathbf{x} - 2\int_S \mathbb{1}\{h_{\mathbf{u}}(\mathbf{x}) \neq y\}\gamma(\mathbf{x})\mathrm{d}\mathbf{x} \\
&= \mathrm{err}_{0-1}^{\mathcal{D}}(h_{\mathbf{u}}) + \mathrm{err}_{0-1}^{\mathcal{D}}(f) - 2\int_S \mathbb{1}\{h_{\mathbf{u}}(\mathbf{x}) \neq y\}\gamma(\mathbf{x})\mathrm{d}\mathbf{x} \; .
\end{aligned}
$$

To finish the proof, note that $\int_S \mathbb{1}\{h_{\mathbf{u}}(\mathbf{x}) \neq y\}\gamma(\mathbf{x})\mathrm{d}\mathbf{x} \geq 0$. $\qquad\square$

### A.3 Proof of Lemma 3.5

*Proof.* Let $R, U$ be the absolute constants from the Definition 1.2. If we set $\rho = \frac{R^2}{32U}$, by Claim 3.4, to guarantee $\mathrm{err}_{0-1}^{\mathcal{D}_{\mathbf{x}}}(h_{\bar{\mathbf{w}}}, f) \leq \sigma$ it suffices to show that the angle $\theta(\bar{\mathbf{w}}, \mathbf{w}^*) \leq O(\sigma) =: \theta_0$. Using (the contrapositive of) Lemma 3.2, if the norm squared of the gradient of some vector $\mathbf{w} \in \mathbb{S}^{d-1}$ is smaller than $\rho$, then $\mathbf{w}$ is close to either $\mathbf{w}^*$ or $-\mathbf{w}^*$ – that is, $\theta(\mathbf{w}, \mathbf{w}^*) \leq \theta_0$ – or $\theta(\mathbf{w}, -\mathbf{w}^*) \leq \theta_0$. Therefore, it suffices to find a point $\mathbf{w}$ with gradient $\|\nabla_{\mathbf{w}}\mathcal{L}_\sigma(\mathbf{w})\|_2 \leq \rho$. From Lemma 3.3, after $T = O(\frac{d}{\sigma^4\rho^4}\log(1/\delta))$ steps, the norm of the gradient of some vector in the list $(\mathbf{w}^{(0)}, \ldots, \mathbf{w}^{(T)})$ will be at most $\rho$ with probability $1 - \delta$. Therefore, the required number of iterations is $T = \mathrm{poly}(U/R) \cdot d\frac{\log(1/\delta)}{\sigma^4}$. Note that one of the hypotheses in the list that is returned by Algorithm 1 is $\sigma$-close to the true $\mathbf{w}^*$. From Claim 3.4, we have that there exists a $\hat{\mathbf{w}} \in L$ such that $\mathrm{err}_{0-1}^{\mathcal{D}}(h_{\hat{\mathbf{w}}}) \leq \mathrm{opt} + O(\sigma) = \mathrm{opt} + O(\sigma)$. $\qquad\square$

### A.4 Proof of Theorem 1.3

*Proof of Theorem 1.3.* Let $R, U$ be the absolute constants from Definition 1.2. and let $C = 2^{15}U^3/R^4$. We will do binary search to find the correct value of $\sigma$ using a grid of size $O(1/\epsilon)$. In particular, we consider $\sigma \in \{C\epsilon, (C+1)\epsilon, \ldots, C\}$. We now analyze our binary search over this grid. We have three cases. We first assume that $\epsilon \leq \mathrm{opt} \leq C$. Let $L_k$ be the list of candidates output by Algorithm 1 for $\sigma = k \cdot \epsilon$. Note that there is a value of $k$ such that $\mathrm{opt} < C\sigma$ and $\mathrm{opt} > C\sigma - \epsilon$. Then we have that there exists $\hat{\mathbf{w}} \in L_k$ such that $\mathrm{err}_{0-1}(h_{\hat{\mathbf{w}}}) \leq \mathrm{opt} + O(\sigma) = O(\mathrm{opt}) + \epsilon$. To find the right value of $k$, we do binary search in the $O(1/\epsilon)$-sized grid of possible values and check each time if we obtained a weight vector that decreased the overall error. Thus, we will overall construct $\mathrm{poly}(R/U) \cdot \log(1/\epsilon)$ lists. Finally, to evaluate all the vectors from the list, we need a small number of samples from the distribution $\mathcal{D}$ to obtain the best among them, i.e., the one that minimizes the zero-one loss. The maximum size of each list of candidates is $\mathrm{poly}(U/R) \cdot d\frac{\log(1/\delta)}{\epsilon^4}$, Therefore, from Hoeffding's inequality, it follows that $O(\log(d/(\epsilon\delta))/\epsilon^2)$ samples are sufficient to guarantee that the excess error of the chosen hypothesis is at most $\epsilon$ with probability at least $1 - \delta$. Similarly, in the case where $\mathrm{opt} \leq \epsilon$ we have that for $\sigma = C\epsilon$, by running Algorithm 1, we obtain a list $L_1$ of candidates. From Lemma 3.5, we get that there is a vector $\hat{\mathbf{w}} \in L_1$, such that $\mathrm{err}_{0-1}(h_{\hat{\mathbf{w}}}) \leq \mathrm{opt} + O(\sigma) \leq O(\epsilon)$. If $\mathrm{opt} \geq C$ then any halfspace will have error $\mathrm{err}_{0-1}(h_{\hat{\mathbf{w}}}) \leq \mathrm{poly}(R/U) = O(\mathrm{opt})$. We conclude that the total number of samples will be $\widetilde{O}(d\log(1/\delta)/\epsilon^4)$. This completes the proof. $\qquad\square$

## B Omitted Proofs from Section 4

In this section, we show that optimizing convex surrogates of the zero-one loss cannot get error $O(\mathrm{opt}) + \epsilon$, even under Gaussian marginals. Recall that we consider objectives of the form

$$
\mathcal{C}(\mathbf{w}) = \mathop{\mathbf{E}}_{\mathbf{x},y\sim\mathcal{D}}[\ell(-y\langle\mathbf{x}, \mathbf{w}\rangle)] \,, \tag{12}
$$

where $\ell(\cdot)$ is a convex loss function. Notice that by considering the *population* version of the objective in Equation (2), we essentially rule out the possibility of sampling errors to be the reason that the minimizer of the convex objective did not achieve low classification error. With standard tools from empirical processes, one can readily get the same result for the empirical objective

443  $(1/N) \sum_{i=1}^{N} \ell(-y^{(i)} \langle \mathbf{x}^{(i)}, \mathbf{w} \rangle)$ assuming that the sample size $N$ is sufficiently large. We now restate
444  the main result of this section that allows us to show Theorem 1.4.

445  **Theorem B.1.** *Fix $Z > 0, \theta \in (0, \pi/8)$, and let $\mathcal{D}_{\mathbf{x}}$ be a radially symmetric distribution on $\mathbb{R}^2$ such*
446  *that*

447    *1. For all $t > 0$ it holds $\mathbf{Pr}_{\mathbf{x} \sim \mathcal{D}_{\mathbf{x}}}[\|\mathbf{x}\|_2 \geq t] > 0$.*

448    *2. $\mathbf{E}_{\mathbf{x} \sim \mathcal{D}_{\mathbf{x}}} [\mathbb{1}\{\|\mathbf{x}\|_2 \geq Z\} \|\mathbf{x}\|_2] > 24\theta \; \mathbf{E}_{\mathbf{x} \sim \mathcal{D}_{\mathbf{x}}} [\|\mathbf{x}\|_2].$*

449  *Then there exists a distribution $\mathcal{D}$ on $\mathbb{R}^2 \times \{\pm 1\}$ and a halfspace $\mathbf{w}^*$ such that $\mathrm{err}_{0-1}^{\mathcal{D}}(\mathbf{w}^*) \leq$*
450  *$\mathbf{Pr}_{\mathbf{x} \sim \mathcal{D}_{\mathbf{x}}}[\|\mathbf{x}\|_2 \geq Z]$, the $\mathbf{x}$-marginal of $\mathcal{D}$ is $\mathcal{D}_{\mathbf{x}}$, and for every convex, non-decreasing, non-constant*
451  *loss $\ell(\cdot)$ and every $\mathbf{w}$ such that $\theta(\mathbf{w}, \mathbf{w}^*) \leq \theta$ it holds $\nabla_{\mathbf{w}} \mathcal{C}(\mathbf{w}) \neq \mathbf{0}$, where $\mathcal{C}$ is defined in Eq. (2).*

452  *Proof.* We start by constructing the noisy distribution $\mathcal{D}$. Fix any unit vector $\mathbf{w}^*$ and let $\widetilde{\mathbf{w}}$ be a vector
453  such that $\theta(\mathbf{w}^*, \widetilde{\mathbf{w}}) = \theta_2$, where $2\theta \leq \theta_2 \leq \pi/4$. Denote by $\widetilde{\mathbf{w}}^\perp$ the vector that is perpendicular
454  with $\widetilde{\mathbf{w}}$ and satisfies $\langle \mathbf{w}^*, \widetilde{\mathbf{w}}^\perp \rangle \geq 0$. We now define the regions $C, S$ that will help us define the parts
455  of the distribution where we will introduce noise by flipping the $y$-labels, see also Figure 1.

$$C = \left\{ \mathbf{x} : \; \langle \mathbf{w}^*, \mathbf{x} \rangle \langle \widetilde{\mathbf{w}}, \mathbf{x} \rangle \geq 0 \text{ and } \langle \widetilde{\mathbf{w}}^\perp, \mathbf{x} \rangle \leq 0 \right\} \qquad S = \{ \mathbf{x} : \; \|\mathbf{x}\|_2 \geq Z \}.$$

456  We are now ready to define our noisy distribution $\mathcal{D}$: *we flip the labels of all points in the set $S \setminus C$.*
457  Observe that $\mathrm{err}_{0-1}^{\mathcal{D}}(\mathbf{w}^*) \leq \mathbf{Pr}_{\mathbf{x} \sim \mathcal{D}_{\mathbf{x}}}[\|\mathbf{x}\|_2 \geq Z]$. Take any $\mathbf{w}$ such that $\theta_1 = \theta(\mathbf{w}, \mathbf{w}^*) \leq \theta$. We
458  are going to bound from below the norm of the gradient of $\mathcal{C}$ at $\mathbf{w}$. The gradient of $\mathcal{C}(\mathbf{w})$ is

$$\nabla_{\mathbf{w}} \mathcal{C}(\mathbf{w}) = \mathop{\mathbf{E}}_{(\mathbf{x}, y) \sim \mathcal{D}}[-y\mathbf{x} \, \ell'(-y \langle \mathbf{x}, \mathbf{w} \rangle)].$$

459  Without loss of generality, we may assume that $\mathbf{w} = \rho \mathbf{e}_2$, where $\rho = \|\mathbf{w}\|_2 > 0$. We have that the
460  first coordinate of the gradient is

$$\langle \nabla_{\mathbf{w}}(\mathcal{C}(\mathbf{w})), \mathbf{e}_1 \rangle = \mathop{\mathbf{E}}_{(\mathbf{x}, y) \sim \mathcal{D}}[-y\mathbf{x}_1 \, \ell'(-y\rho \, \mathbf{x}_2)]. \tag{13}$$

461  In what follows, we are going to use polar coordinates $(r, \phi)$ with the standard relation to Cartesian
462  $(\mathbf{x}_1, \mathbf{x}_2) = (r \cos \phi, r \sin \phi)$. Now assume that we want to compute the contribution of a specific
463  region $A = \{r \in [r_1, r_2], \phi \in [\phi_1, \phi_2]\}$ to the gradient of Equation (13). We denote the 2-
464  dimensional density of the radially symmetric distribution $\mathcal{D}_{\mathbf{x}}$ as $\gamma(r)$. We have

$$\mathop{\mathbf{E}}_{(\mathbf{x}, y) \sim \mathcal{D}}[-y\mathbf{x}_1 \, \ell'(-y\mathbf{x}_2) \mathbb{1}_A(\mathbf{x})] = \int_{r_1}^{r_2} r\gamma(r) \int_{\phi_1}^{\phi_2} -yr \cos \phi \, \ell'(-y\rho \, r \sin \phi) \mathrm{d}\phi \mathrm{d}r$$

$$= \frac{1}{\rho} \int_{r_1}^{r_2} r\gamma(r) \int_{\phi_1}^{\phi_2} (\ell(-y\rho r \sin \phi))' \mathrm{d}\phi \mathrm{d}r = \frac{1}{\rho} \int_{r_1}^{r_2} r\gamma(r)(\ell(-y\rho r \sin \phi_2) - \ell(-y\rho r \sin \phi_1)) \mathrm{d}r.$$

$$\tag{14}$$

465  Without loss of generality, we consider the two cases shown in Figure 1. We start with the
466  first case, where $\mathbf{w}$ lies between $\mathbf{w}^*$ and $\widetilde{\mathbf{w}}$. We first compute the contribution to the gradi-
467  ent in $S^c$, i.e., the points where $y = \mathrm{sign}(\langle \mathbf{w}^*, \mathbf{x} \rangle)$. Since the distribution is radially symmet-
468  ric, we have $\mathbf{E}_{(\mathbf{x}, y) \sim \mathcal{D}}[-y\mathbf{x}_1 \, \ell'(-y\mathbf{x}_2) \mathbb{1}_{S^c}(\mathbf{x})] = 2 \, \mathbf{E}_{(\mathbf{x}, y) \sim \mathcal{D}}[-y\mathbf{x}_1 \, \ell'(-y\mathbf{x}_2) \mathbb{1}_{R_1}(\mathbf{x})]$, where
469  $R_1 = \{r \in [0, Z], \phi \in [\theta_1, \pi + \theta_1]\}$. From Equation (14), we obtain that

$$I_{S^c} = \mathop{\mathbf{E}}_{(\mathbf{x}, y) \sim \mathcal{D}}[-y\mathbf{x}_1 \, \ell'(-y\mathbf{x}_2) \mathbb{1}_{S^c}(\mathbf{x})] = \frac{2}{\rho} \int_0^Z r\gamma(r)(\ell(\rho r \, \sin \theta_1) - \ell(-\rho r \, \sin \theta_1)) \mathrm{d}r.$$

470  Observe that since $\ell(\cdot)$ is non-decreasing we have $I_{S^c} \geq 0$. Next we compute the contribution
471  of region $S$ to the gradient. Recall that $S$ contains $S \setminus C$, i.e., the region we flipped the labels,
472  $y = -\mathrm{sign}(\langle \mathbf{w}^*, \mathbf{x} \rangle)$, see Figure 1. Using again the fact that the distribution is radially symmetric
473  and Equation (13) for the region $R_2 = \{r \in [Z, +\infty), \phi \in [\pi/2 - \theta_2, 3\pi/2 - \theta_2]\}$, we obtain

$$I_S = \mathop{\mathbf{E}}_{(\mathbf{x}, y) \sim \mathcal{D}}[-y\mathbf{x}_1 \, \ell'(-y\mathbf{x}_2) \mathbb{1}_S(\mathbf{x})] = \frac{2}{\rho} \int_Z^\infty r\gamma(r) \Big( \ell(\rho r \, \sin(\frac{3\pi}{2} - \theta_2)) - \ell(\rho r \, \sin(\frac{\pi}{2} - \theta_2)) \Big) \mathrm{d}r$$

$$= \frac{2}{\rho} \int_Z^\infty r\gamma(r)(\ell(-\rho r \, \cos \theta_2) - \ell(\rho r \, \cos \theta_2)) \mathrm{d}r.$$

474 Similarly to the previous case, the fact that $\ell(\cdot)$ is non-decreasing implies that $I_S \leq 0$.

475 Now we are going to use the facts that $\ell(\cdot)$ is convex and non-decreasing. Since both $\theta_1, \theta_2 \leq \pi/4$,
476 we have that $\cos\theta_2 \geq \sin\theta_1$ and therefore, from the convexity of $\ell(\cdot)$, we obtain

$$\frac{\ell(\rho r \sin(\theta_1)) - \ell(-\rho r \sin\theta_1)}{2\rho r \sin\theta_1} \leq \frac{\ell(\rho r \cos\theta_2) - \ell(-\rho r \sin\theta_1)}{\rho r \cos(\theta_2) + \rho r \sin(\theta_1)} \ .$$

477 Since $\ell(\cdot)$ is also non-decreasing, we have that $\ell(\rho r \cos\theta_2) - \ell(-\rho r \sin\theta_1) \leq \ell(\rho r \cos\theta_2) -$
478 $\ell(-\rho r \cos\theta_2)$ and therefore,

$$\ell(\rho r \sin\theta_1) - \ell(-\rho r \sin\theta_1) \leq \frac{2\sin\theta_1}{\cos\theta_2 + \sin\theta_1}\left(\ell(\rho r \cos\theta_2) - \ell(-\rho r \cos\theta_2)\right) \ .$$

479 To simplify notation, we define the functions $\bar{\ell}(r) = \ell(\rho r \cos\theta_2)$ and $h(r) = \bar{\ell}(r) - \bar{\ell}(-r)$. Observe
480 that $\bar{\ell}(\cdot)$ enjoys exactly the same properties as $\ell(\cdot)$, that is $\bar{\ell}(\cdot)$ is convex, non-decreasing, and
481 non-constant. Moreover, observe that $h(r)$ is non-negative and non-decreasing. Using the above
482 inequalities, we obtain that

$$\rho\left\langle\nabla_{\mathbf{w}}\mathcal{C}(\mathbf{w}), \mathbf{e}_1\right\rangle = \rho(I_S + I_{S^c}) \leq \frac{4\sin\theta_1}{\cos\theta_2 + \sin\theta_1}\underbrace{\int_0^Z r\gamma(r)h(r)\mathrm{d}r}_{I_2} - 2\underbrace{\int_Z^\infty r\gamma(r)h(r)\mathrm{d}r}_{I_1} \ . \quad (15)$$

483 We will now show that instead of dealing with every convex and increasing $\bar{\ell}(\cdot)$, we can restrict our
484 attention to simple piecewise-linear convex and increasing functions. First, we observe that without
485 loss of generality we may assume that the convex function $\bar{\ell}(r)$ is constant for all $r \leq -Z$, since
486 that part only increases $I_1$. To construct $s(\cdot)$, we use the supporting lines of $\bar{\ell}(\cdot)$ at $-Z$ and $0$, and
487 the secant line from $0$ to $Z$. We will first assume that $\bar{\ell}'(Z) > 0$. Let $a_0$ be a subgradient of $\bar{\ell}(\cdot)$ at
488 $0$. Then the secant from $0$ to $Z$ is some line $a_1 r - a_0 Z_0$ for some $a_1 \in [a_0, \bar{\ell}'(Z)]$. Then, for every
489 convex and non-decreasing $\bar{\ell}(\cdot)$, the following piecewise-linear function $s(r)$ makes the ratio $I_1/I_2$
490 smaller. In what follows, $Z_0 \in [-Z, 0]$ is the intersection point of the supporting line $a_0 r - a_0 Z_0$
491 and the constant supporting line at $-Z$.

$$s(r) = b + \begin{cases} 0, & r \leq Z_0 \\ a_0 r - a_0 Z_0, & Z_0 < r \leq 0 \\ a_1 r - a_0 Z_0, & 0 < r \end{cases} \ .$$

492 We have

$$h(r) = \begin{cases} (a_1 + a_0)r, & 0 \leq r \leq -Z_0, \\ a_1 r - a_0 Z_0 & -Z_0 < r \end{cases} \ .$$

493

$$I_1 = a_1 \int_Z^\infty r^2\gamma(r)dr - a_0 Z_0 \int_Z^\infty r\gamma(r)dr \geq a_1 \int_Z^\infty r^2\gamma(r)dr \ .$$

494

$$I_2 = (a_1 + a_0)\int_0^{-Z_0} r^2\gamma(r)dr + a_1 \int_{-Z_0}^Z r^2\gamma(r)dr - a_0 Z_0 \int_{-Z_0}^Z r\gamma(r)dr$$

$$\leq 2(a_1 + a_0)\int_0^Z r^2\gamma(r)dr \leq 4a_1 \int_0^Z r^2\gamma(r)dr \ .$$

495 Using the above bounds in Equation (15), we obtain

$$\left\langle\nabla_{\mathbf{w}}\mathcal{C}(\mathbf{w}), \mathbf{e}_1\right\rangle \leq \frac{2a_1}{\rho}\left(\frac{8\sin\theta_1}{\cos\theta_2 + \sin\theta_1}\int_0^Z r^2\gamma(r)\mathrm{d}r - \int_Z^\infty r^2\gamma(r)\mathrm{d}r\right) \ .$$

496 Removing the positive quantity $\sin\theta_1$ of the denominator and replacing $\theta_1$ by its upper bound $\theta$, we
497 obtain the claimed bound. Since $\cos\theta_2$ is decreasing in $[0, \pi/2]$, we may choose $\theta_2 = 2\theta$. Our final
498 bound is then

$$\left\langle\nabla_{\mathbf{w}}\mathcal{C}(\mathbf{w}), \mathbf{e}_1\right\rangle \leq \frac{2a_1}{\rho}\left(8\tan(2\theta)\int_0^Z r^2\gamma(r)\mathrm{d}r - \int_Z^\infty r^2\gamma(r)\mathrm{d}r\right)$$

$$\leq \frac{2a_1}{\rho}\left(24\theta \mathop{\mathbf{E}}_{\mathbf{x}\sim\mathcal{D}_{\mathbf{x}}}[\|\mathbf{x}\|_2] - \mathop{\mathbf{E}}_{\mathbf{x}\sim\mathcal{D}_{\mathbf{x}}}[\mathbb{1}\{\|\mathbf{x}\|_2 > Z\}\|\mathbf{x}\|_2]\right) \ ,$$

499 where for the last inequality we used the fact that $\tan(2\theta) \leq 3\theta$ for all $\theta \in [0, \pi/8]$. In the case
500 where $\ell'(\rho Z \cos\theta_2) = 0$, the above bound vanishes. We fist assume that this is not the case. Then,
501 using Assumption 2 of our theorem, we obtain that $\langle \nabla_{\mathbf{w}} \mathcal{C}(\mathbf{w}), \mathbf{e}_1 \rangle \neq 0$ and therefore $\nabla_{\mathbf{w}} \mathcal{C}(\mathbf{w}) \neq \mathbf{0}$.

502 In the case where $\ell'(\rho Z \cos\theta_2) = 0$, we observe that $I_{S^c}$ vanishes. To finish the proof, we need to
503 bound from above and away from zero the integral $I_S$. Since $\bar{\ell}(\cdot)$ is non-constant, there exists a point
504 $Z' > Z$ with $\bar{\ell}'(Z) > 0$. Convexity of $\bar{\ell}(\cdot)$ implies $h(r) \geq \bar{\ell}'(Z)r$. Using this fact, we get

$$I_S \leq -\bar{\ell}'(Z') \int_{Z'}^{\infty} r^2 \gamma(r) \mathrm{d}r \,.$$

505 Using Assumption 1 of our theorem, we again get that $\nabla_{\mathbf{w}} \mathcal{C}(\mathbf{w}) \neq \mathbf{0}$.

506 Next we handle the case where the candidate $\mathbf{w}$ lies out of the cone formed by $\mathbf{w}^*$ and $\widetilde{\mathbf{w}}$. In that
507 case, similarly to before, we compute the contribution to the gradient of the noisy samples $S$ and the
508 non-noisy $S^c$.

$$I_{S^c} = \mathop{\mathbf{E}}_{(\mathbf{x},y)\sim\mathcal{D}}[-y\mathbf{x}_1\,\ell'(-y\mathbf{x}_2)\mathbb{1}_{S^c}(\mathbf{x})] = \frac{2}{\rho}\int_0^Z r\gamma(r)(\ell(-\rho r\,\sin\theta_1) - \ell(\rho r\,\sin\theta_1)\mathrm{d}r \,.$$

509 and

$$I_S = \mathop{\mathbf{E}}_{(\mathbf{x},y)\sim\mathcal{D}}[-y\mathbf{x}_1\,\ell'(-y\mathbf{x}_2)\mathbb{1}_S(\mathbf{x})] = \frac{2}{\rho}\int_Z^{\infty} r\gamma(r)(\ell(-\rho r\,\cos\theta_2) - \ell(\rho r\,\cos\theta_2)\mathrm{d}r \,.$$

510 In contrast to the previous case, we now observe that since $\ell(\cdot)$ is non-decreasing, both $I_S$ and $I_{S^c}$
511 have the same sign, i.e., they are both non-positive. From Assumption 1, and the fact that $\ell(\cdot)$ is
512 non-constant, we obtain that $I_S + I_{S^c} < 0$, which in turn implies that $\nabla_{\mathbf{w}} \mathcal{C}(\mathbf{w}) \neq \mathbf{0}$. $\qquad\square$

513 We are now ready to give the proof of Theorem 1.4, which we restate below for convenience.

514 **Theorem 1.4.** Let $\mathcal{D}_{\mathbf{x}}$ be the standard normal distribution on $\mathbb{R}^d$. There exists a distribu-
515 tion $\mathcal{D}$ on $\mathbb{R}^d \times \{\pm 1\}$ such that for *every* convex, non-decreasing loss $\ell(\cdot)$, the objective
516 $\mathcal{C}(\mathbf{w}) = \mathbf{E}_{\mathbf{x},y\sim\mathcal{D}}[\ell(-y\langle\mathbf{x},\mathbf{w}\rangle)]$ is minimized at some halfspace $h$ with error $\mathrm{err}_{0-1}^{\mathcal{D}}(h) =$
517 $\Omega(\mathrm{opt}\sqrt{\log(1/\mathrm{opt})})$. Moreover, there exists a log-concave marginal $\mathcal{D}_{\mathbf{x}}$ (resp. $s$-heavy tailed
518 marginal) such that $\mathrm{err}_{0-1}^{\mathcal{D}}(h) = \Omega(\mathrm{opt}\log(1/\mathrm{opt}))$ (resp. $\mathrm{err}_{0-1}^{\mathcal{D}}(h) = \Omega(\mathrm{opt}^{1-1/s})$).

519 *Proof.* Since all the examples that we are going to consider will be radially invariant distributions,
520 we note that the "disagreement" error of two halfspaces with normal vectors $\mathbf{v}, \mathbf{u}$ is $\theta(\mathbf{v},\mathbf{u})/\pi$.
521 From Claim 3.4, we have that the classification error of any candidate $\mathbf{w}$ is lower bounded by
522 $\theta(\mathbf{w},\mathbf{w}^*)/\pi - \mathrm{opt}$. We will construct a distribution $\mathcal{D}$ such that there is some $\mathbf{w}^*$ that achieves error
523 opt, but at the same time $\mathcal{C}(\mathbf{w})$ is minimized at some halfspace such that $\theta(\mathbf{w},\mathbf{w}^*) = \omega(\mathrm{opt})$. This
524 means that the minimizer of $\mathcal{C}$ has classification error $\omega(\mathrm{opt})$.

525 We assume first that $\mathcal{D}_{\mathbf{x}}$ is the standard normal and without loss of generality work in two dimensions.
526 Recall that the density function in this case is radially invariant, i.e., $\gamma(\mathbf{x}_1,\mathbf{x}_2) = \frac{1}{2\pi}e^{-\|\mathbf{x}\|_2^2/2}$. If $\ell$
527 is a constant function, any halfspace would minimize it and therefore, this case is trivial. Clearly,
528 Assumption 1 of Theorem B.1 holds in this case. We now show that we can pick $Z > 0$ such that the
529 probability of all points with flipped label is $O(\mathrm{opt})$ and make Assumption 2 of Theorem B.1 true.
530 Since the distribution is Gaussian, we have that for $Z = \Theta(\sqrt{\log(1/\mathrm{opt})})$ it holds $\Pr[\|\mathbf{x}\|_2 \geq Z] \leq$
531 opt. Since the distribution is isotropic, we have $\mathbf{E}_{\mathbf{x}\sim\mathcal{D}_{\mathbf{x}}}[\|\mathbf{x}\|_2] \leq \sqrt{\mathbf{E}_{\mathbf{x}\sim\mathcal{D}_{\mathbf{x}}}[\|\mathbf{x}\|_2^2]} = 1$. Moreover,
532 we have that

$$\mathop{\mathbf{E}}_{\mathbf{x}\sim\mathcal{D}_{\mathbf{x}}}[\mathbb{1}\{\|\mathbf{x}\|_2 \geq Z\}\|\mathbf{x}\|_2] = \int_Z^{\infty} r^2 e^{-r^2/2}\mathrm{d}r \geq e^{-Z^2/2}Z = \Theta(\mathrm{opt}\sqrt{\log(1/\mathrm{opt})}) \,.$$

533 Now we can fix some $\theta = \Omega(\mathrm{opt}\sqrt{\log(1/\mathrm{opt})}) < \pi/8$ and observe that Assumption 2 of The-
534 orem B.1 is satisfied. Therefore, we have that for any halfspace with normal vector $\mathbf{w}$ with
535 $\theta(\mathbf{w},\mathbf{w}^*) \leq \theta = \Omega(\mathrm{opt}\sqrt{\log(1/\mathrm{opt})})$ it holds that $\nabla_{\mathbf{w}}\mathcal{C}(\mathbf{w}) \neq \mathbf{0}$, and therefore it cannot be a
536 minimizer of $\mathcal{C}(\mathbf{w})$.

537 For the log-concave marginals the argument is similar. We work again in two dimensions and pick
538 $\gamma(\mathbf{x}) = \frac{6}{\pi}e^{-2\sqrt{3}\|\mathbf{x}\|_2}$. This distribution is isotropic log-concave. We have that for $Z = \Theta(\log(1/\mathrm{opt}))$

539    it holds that $\mathbf{Pr}[\|\mathbf{x}\|_2 \geq Z] \leq \mathrm{opt}$. Moreover, we have $\mathbf{E}_{\mathbf{x} \sim \mathcal{D}_{\mathbf{x}}}\left[\mathbb{1}\{\|\mathbf{x}\|_2 \geq Z\} \|\mathbf{x}\|_2\right] \geq$
540    $(\sqrt{3}/(2\pi))e^{-2\sqrt{3}Z}Z = \Omega(\mathrm{opt}\log(1/\mathrm{opt}))$.

541    Now we can fix some $\theta = \Omega(\mathrm{opt}\log(1/\mathrm{opt})) < \pi/8$ and observe that Assumption 2 of Theorem B.1
542    is satisfied. Therefore, we have that for any halfspace with normal vector $\mathbf{w}$ with $\theta(\mathbf{w}, \mathbf{w}^*) \leq \theta = $
543    $\Omega(\mathrm{opt}\log(1/\mathrm{opt}))$ it holds that $\nabla_{\mathbf{w}}\mathcal{C}(\mathbf{w}) \neq \mathbf{0}$, and as a result it cannot be a minimizer of $\mathcal{C}(\mathbf{w})$.

544    For the heavy tailed marginals, the argument is similar. We work again in two dimensions, and for
545    any $s > 2$ we pick

$$\gamma(\mathbf{x}) = \frac{b_s}{\left(\frac{\|\mathbf{x}\|_2}{a_s} + 1\right)^{2+s}} \,,$$

546    where the constants $a_s, b_s$ depend only on $s > 2$ and are appropriately picked so that the distribution
547    is isotropic. Using polar coordinates, we have

$$\mathbf{Pr}[\|\mathbf{x}\|_2 \geq Z] = 2\pi \int_Z^{\infty} \frac{r b_s}{\left(\frac{r}{a_s} + 1\right)^{2+s}} \mathrm{d}r = \frac{2\pi b_s}{s(1+s)} \frac{a_s + (s+1)Z}{(a_s + Z)^{1+s}} \,.$$

548    Therefore, for $Z = \Theta((1/\mathrm{opt})^{1/s})$ it holds that $\mathbf{Pr}[\|\mathbf{x}\|_2 \geq Z] \leq \mathrm{opt}$. Moreover, we have

$$\mathop{\mathbf{E}}_{\mathbf{x} \sim \mathcal{D}_{\mathbf{x}}}\left[\mathbb{1}\{\|\mathbf{x}\|_2 \geq Z\} \|\mathbf{x}\|_2\right] = 2\pi \int_Z^{\infty} \frac{r^2 b_s}{\left(\frac{r}{a_s} + 1\right)^{2+s}} \mathrm{d}r = \frac{b_s\left(2a_s^2 + 2a_s(s+1)Z + s(s+1)Z^2\right)}{s\left(s^2 - 1\right)(a_s + Z)^{s+1}} \,.$$

549    Therefore, for $Z = \Theta((1/\mathrm{opt})^{1/s})$ it holds $\mathbf{E}_{\mathbf{x} \sim \mathcal{D}_{\mathbf{x}}}\left[\mathbb{1}\{\|\mathbf{x}\|_2 \geq Z\} \|\mathbf{x}\|_2\right] = \Omega(\mathrm{opt}^{1-1/s})$. We can
550    now fix some $\theta = \Omega(\mathrm{opt}^{1-1/s}) < \pi/8$ and observe that Assumption 2 of Theorem B.1 is satisfied.
551    Therefore, we have that for any halfspace with normal vector $\mathbf{w}$ with $\theta(\mathbf{w}, \mathbf{w}^*) \leq \theta = \Omega(\mathrm{opt}^{1-1/s})$
552    it holds that $\nabla_{\mathbf{w}}\mathcal{C}(\mathbf{w}) \neq \mathbf{0}$, and as a result it cannot be a minimizer of $\mathcal{C}(\mathbf{w})$.     $\square$