[Reviews · NeurIPS 2020]

Review 1

Summary and Contributions: The paper studies the fundamental problem of learning halfspaces under adversarial noise. It is shown that by running projected SGD on a non-convex function, one is always guaranteed with an near-optimal halfspace.

Strengths: - This work is technically sound. It gives a very simple formulation, i.e. minimizing a properly scaled sigmoid function, and shows that any stationary point serves as a good approximation to the desired halfspace. - Although similar idea has been explored in a very recent work for Massart noise, authors clarified that adversarial noise is a more involved model and requires extra treatments, e.g. the scaling parameter in the objective function cannot be too small. - The theoretical guarantee holds also for a family of heavy-tailed marginal distributions. This might be of independent interest.

Weaknesses: I like this paper but couple of concerns needs to be addressed. - The comparison to [ABL17] is not convincing to me. I do not really agree that [ABL17] is not practical. First of all, their algorithm runs in log(1/eps) epochs, and thus has an exponentially better dependence on 'eps' than this work, in terms of label complexity. Second, in each epoch, they optimize a convex program up to a constant accuracy. Therefore, the per-epoch computational complexity is very mild. - Technical contribution compared to the Massart noise [DKTZ20] needs more clarification. While I agree with the authors that adversarial noise is a different type from Massart noise, some prior works already showed that from the technical perspective, there might a towards unified way to analyze them. For example, in [A], it was shown that for both noise models, most of the analysis follows the same way: one only needs to slightly treat a concentration inequality for each. See Lemma 21 and Lemma 22 therein. My overall feeling is that this work presents a simpler formulation (which is appreciated), achieving similar excess error bound to [ABL17], but has worse label complexity, which makes the work not as exciting as [DKTZ20]. My another concern is that the technical contribution might be marginal in light of the published work of [DKTZ20]. I am happy to hear the author's response and reconsider the rating if they address my concerns. [A] Efficient active learning of sparse halfspaces, C. Zhang, COLT 2018 ---updates after author rebuttal--- Thank you for your response. It addressed some of my concerns. However, I still feel the contribution is marginal.

Correctness: Yes.

Clarity: Yes

Relation to Prior Work: Yes.

Reproducibility: Yes

Additional Feedback:


Review 2

Summary and Contributions: The paper is about agnostically learning homogeneous halfspaces in the distribution specific PAC model with adversarial label noise. It shows that SGD on a non-convex surrogate of the zero-one loss solves the problem with $O(opt) + \epsilon$ error in poly time (in d and \epsilon), where *opt* is the misclassification error of the best hyperplane. Interestingly, the authors also provide a lower bound result of showing that optimizing any convex surrogate will incur $\Omega(opt)$ error. Combining the first result, it means that applying SGD to the non-convex surrogate might be a better approach for solving the problem, compared to applying SGD to the convex surrogates.

Strengths: While there are some previous works studying the same problem (e.g. [Awasthi et al. 2017], the algorithms in those works seem to be complicated. One of the contributions of this paper is showing that standard SGD on a non-convex objective works for the problem, which is by building on the result of [Diakonikolas et al. 2020]. The *separateness* result between the non-convex surrogate and convex surrogates is novel and interesting. Overall, I think the contributions are significant and the proof/analysis seems to be correct.

Weaknesses: I understand that this is a theory paper. But it will be better if a proof-of-concept experiment is provided.

Correctness: Yes.

Clarity: Yes. But it seems to me that on line 419-422 and line 432 in the supplementary, you overload the notation 'C'. It looks like you use C to represent a quantity and its inverse simultaneously.

Relation to Prior Work: Yes, detailed comparison with the prior works is provided.

Reproducibility: Yes

Additional Feedback: *** after rebuttal *** I've read the authors' rebuttal.


Review 3

Summary and Contributions: The paper studies the problem of agnostic learning of homogeneous halfspaces under structured distributions. For this problem, there exists algorithms that achieve O(opt) + epsilon error in time poly(d, 1/eps), where d=dim. of data, opt = error of best homogeneous halfspace and eps is an error tolerance. These algorithms are typically iterative in nature and in each iteration minimize a convex function. The current paper shows that the same results can be obtained via SGD on a non-convex objective. The paper builds upon a recent work of [DKTZ20] to show that a smooth approximation to the 0-1 loss based on a logistic form suffices. In particular, the main claim is Lemma 3.2 showing that stationary points of the loss function will be close to the true halfspaces, and hence will have small error. *******Post Rebuttal******** I've read the authors' rebuttal and my score remains unchanged.

Strengths: + shows that a recent approach designed for benign noise models such as the Massart noise models also works for agnostic learning.

Weaknesses: - Relies heavily on [DKTZ20]. Claim 3.4 is fairly standard and Lemma 3.2 essentially uses the analysis of [DKTZ20]. - It is unclear of the proposed algorithm is truly simpler since one needs to do a grid search on the value of opt (in order to set sigma).

Correctness: Yes, the claims are correct.

Clarity: Yes, the paper is largely well written.

Relation to Prior Work: Yes, related work is adequately discussed. However, as mentioned before, the techniques heavily overlap with [DKTZ20].

Reproducibility: Yes

Additional Feedback: The proof of Lemma 3.2 should be modified a bit since the noisy region S is a random set (since the label conditioned on x can also be a random variable).


Review 4

Summary and Contributions: This paper proof that projected SGD on a non-convex loss function of homogeneous halfspace problem converges to a suboptimal solution, where the suboptimality gap is at most O(opt) and opt is the minimum of the loss. By proposing the novel non-convex loss function, the result of the paper simplifies previous algorithms on the aforementioned problem. The paper also shows that the proposed non-convex loss function outperforms all convex and non-decreasing loss function on the order of opt. The reviewer finds the result of the paper interesting and can provide inspiration to the community of optimization research.

Strengths: The paper shows that simple approach (PSGD) with a good choice of loss function, which is non-convex, will converge to a globally good solution in the problem of homogeneous halfspace. The result holds for a variety of distributions and the proof is solid.

Weaknesses: The challenge of analyzing convergence of SGD due to non-convexity is usually in the cases where the hypothesis h(x) is non-convex, e.g., a neural network, which implies that the inspiration of this paper on the study of gradient-base optimization methods in machine learning is limited.

Correctness: I believe the proof is correct, while I did not check all of them.

Clarity: The paper is well written and easy to follow.

Relation to Prior Work: The comparison with previous algorithm on the problem studied is not very clear.

Reproducibility: Yes

Additional Feedback: I hope the author can provide more explanantions on the reason why the non-convex loss is better than other convex loss function.

[Author Response · NeurIPS 2020]

We thank the reviewers for their consideration of our paper and their positive feedback. We start with a summary of
our contributions that we believe addresses several reviewer comments. The main contribution of our work is a strong
separation between convex and non-convex surrogates for the problem of learning halfspaces with adversarial label
noise. On the negative side, we prove that optimizing *any* convex surrogate objective leads to significantly suboptimal
error guarantees. On the positive side, we show that *any* stationary point of a carefully chosen non-convex objective
suffices to obtain nearly best possible error guarantees.

We emphasize that our lower bound result for convex surrogates is a significant contribution of our work, which we
believe is of independent interest. In particular, it applies to most common methods used in practice for classification like
logistic regression and hinge-loss minimization. We show that such **convex optimization approaches are provably**
**suboptimal and simple non-convex methods outperform them**. Based on the reviews, we realize that this message
was lost in our write-up and we will highlight it in the final version.

Our positive structural result (i.e., that any stationary point of our non-convex surrogate works) provides a novel
understanding of the underlying learning problem from an optimization point of view. An immediate corollary is that
vanilla projected SGD (or *any* other method that converges to a stationary point) efficiently solves our learning problem
to near-optimal accuracy. As a result, we also obtain the following implications for free: (1) We give an algorithm with
near-linear sample complexity in the dimension $d$ (which is information-theoretically optimal) that runs in sample-linear
time. Moreover, our algorithm works with a single pass over the data and has minimal memory requirements (we just
need to store one sample at each step). (2) Our algorithm succeeds for a broad class of distributions, including ones
for which no polynomial-time algorithm was previously known. Designing noise-tolerant algorithms under natural
distributional assumptions was stated as an open problem in Section 5 of [ABL17].

**Comparison to [DKTZ20].** While the presentation/structure of our technical section resembles [DKTZ20], the proof
of our positive structural result is very different from that in [DKTZ20]. In terms of similarities, both works establish
that any stationary point of a certain non-convex surrogate works for the corresponding learning problem. Moreover,
both works use a parametric class of functions that are smooth surrogates of the 0-1 loss with a smoothing parameter $\sigma$.

Here we handle adversarial label noise which is a much more challenging noise model than the bounded/Massart model
in [DKTZ20]. The worst-case strategy of the adversary (in the adversarial label model) differs from that in the Massart
model. In particular, one needs to understand how the adversary can distribute the corrupted labels in order to bound
their effect and as a result, our analysis departs significantly from that of [DKTZ20]. Our key technical contribution is
that there exists a value of the smoothing parameter $\sigma$ that works. In [DKTZ20], it was shown that any sufficiently
small value of $\sigma$ suffices, which does not hold in the agnostic model. Here we have to carefully choose the correct value
of $\sigma$. Finally, we remark that to obtain our structural result (Lemma 3.2) we inherently require different distributional
assumptions compared to [DKTZ20] (that include log-concave and certain heavy-tailed distributions).

**Reviewer 1** Our work studies the standard ("passive") PAC learning model with adversarial label noise. A discus-
sion/comparison of label complexity (aka "active learning") is orthogonal to the focus of our submission. In the standard
PAC model with adversarial label noise, our algorithm is simpler, more sample-efficient and significantly faster than
the localization-based method introduced in [ABL17]. [ABL17] establish polynomial upper bounds on the sample
complexity and runtime of their algorithm, but the degree of the polynomial is not specified. As explained in the
preceding discussion, our approach has some high-level similarities with [DKTZ20], but our main technical contribution
(Lemma 3.2) requires a conceptually different proof and different set of assumptions. In particular, there appears to be
no black-box way to reduce our setting to the Massart setting.

**Reviewer 2** We agree that using this notation in lines 419-422 is confusing. We will fix this in the final version.

**Reviewer 3** We note that in the adversarial label noise model, the parameter OPT (i.e., the fraction of corrupted
labels), or a reasonable upper bound, is typically known a priori to the algorithm. Previous algorithms (and in particular
[ABL17]) operate under this assumption. Under such an assumption, our algorithm does not need to "guess" OPT and
amounts to a simple SGD. Moreover, even with this additional guessing step, the sample complexity and runtime of our
algorithm remains near-optimal.

**Reviewer 4** While we study linear classification, our learning problem is non-convex due to the noise in the labels. A
broad range of prior works have used convex surrogates of the zero-one loss to address this non-convexity. In this work,
we show that convex surrogates inherently fail with adversarial label noise. In contrast, we design a simple non-convex
surrogate that we show leads to near-optimal accuracy. The intuition is that in our setting the non-convex landscape is
well-behaved in the sense that *any* stationary point suffices. We show that (1) one should not use convex surrogates to
learn linear classifiers in the presence of noise, and (2) one can replace convex losses with a simple non-convex loss and
get much better classification error. Regarding (1): At a high level, convex loss functions will assign large weights to
samples that are far from the origin. An adversary can take advantage of these points, flip their labels, and make the
error of the convex minimizer large.

[Meta-Review · NeurIPS 2020]

It is shown that running SGD on a non-convex surrogate for the 0-1 loss converges to a near optimal halfspace under adversarial label noise. The resulting algorithm is much simpler than others available in the literature. It is also shown (by a novel lower bound) that using convex surrogates cannot really improve the performance. The reviewers are generally positive about the paper. In the revised version the authors should highlight better the difference compared to [DKTZ20].